# New insights into the drainage of inundated ice-wedge polygons using fundamental hydrologic principles

Dylan R Harp[1], Vitaly Zlotnik[2], Charles J Abolt[1], Bob Busey[3], Sofia T Avendaño[1], Brent D Newman[1], Adam L Atchley[1], Elchin Jafarov[1], Cathy J Wilson[1], and Katrina E Bennett[1]

[1]Earth and Environmental Sciences Division, Los Alamos National Laboratory, Los Alamos, NM, 87544
[2]Earth and Atmospheric Sciences Department, University of Nebraska, Lincoln, NE, 68588-0340
[3]International Arctic Research Center, University of Alaska, Fairbanks, AK, 99775

**Correspondence:** Dylan Harp (dharp@lanl.gov)

**Abstract.** The pathways and timing of drainage from the inundated centers of ice-wedge polygons in a warming climate have important implications for carbon flushing, advective heat transport, and transitions from methane to carbon dioxide dominated emissions. Here, we expand on previous research using a recently developed analytical model of drainage from a low-centered polygon. Specifically, we perform (1) a calibration to field data identifying necessary model refinements and (2) a rigorous model sensitivity analysis that expands on previously published indications of polygon drainage characteristics. This research provides intuition on inundated polygon drainage by presenting the first in-depth analysis of drainage within a polygon based on hydrogeological first principles. We verify a recently developed analytical solution of polygon drainage through a calibration to a season of field measurements. Due to the parsimony of the model, providing the potential that it could fail, we identify the minimum necessary refinements that allow the model to match water levels measured in a low-centered polygon. We find that (1) the measured precipitation must be increased by a factor of around 2.2 and (2) the vertical soil hydraulic conductivity must decrease with increasing thaw depth. Model refinement (1) accounts for runoff from rims into the ice-wedge polygon pond during precipitation events and possible rain gauge under catch and refinement (2) accounts for the decreasing permeability of deeper soil layers. The calibration to field measurements supports the validity of the model indicating that it is able to represent ice-wedge polygon drainage dynamics. We then use the analytical solution in non-dimensional form to provide a baseline for the effects of polygon aspect ratios (radius to thaw depth) and coefficient of hydraulic conductivity anisotropy (horizontal to vertical hydraulic conductivity) on drainage pathways and temporal depletion of ponded water from inundated ice-wedge polygon centers. By varying the polygon aspect ratio, we evaluate the relative effect of polygon size (width), inter-annual increases in active layer thickness, and seasonal increases in thaw depth on drainage. The results of our sensitivity analysis rigorously confirm a previous analysis indicating that most drainage through the active layer occurs along an annular region of the polygon center near the rims. This has important implications for transport of nutrients (such as dissolved organic carbon) and advection of heat towards ice-wedge tops. We also provide a comprehensive investigation of the effect of polygon aspect ratio and anisotropy on drainage timing and patterns expanding on previously published research. Our results indicate that polygons with large aspect ratios and high anisotropy will have the most distributed drainage, while polygons with large aspect ratios and low anisotropy will have their drainage most focused near their periphery and will drain most slowly. Polygons with

small aspect ratios and high anisotropy will drain most quickly. These results, based on parametric investigation of idealized scenarios, provide a baseline for further research considering the geometric and hydraulic complexities of ice-wedge polygons.

*Copyright statement.* TEXT

## 1  Introduction

Polygonal tundra occurs in continuous permafrost landscapes lacking exposed bedrock or active sedimentation (Brown et al.,
1997; Mackay, 2000). Estimates of its spatial extent vary from around 250,000 km$^2$ (Minke et al., 2007), or approximately the size of England , to 2.6 million km$^2$, or approximately 30% of the Arctic land surface (Mackay, 1972). This terrain is rich in organic carbon (Tarnocai et al., 2009; Hugelius et al., 2014) which is poised for release to the atmosphere as carbon dioxide or methane under a warming climate, and may cause a significant feedback to climate change (Schuur et al., 2008). Depressions in the microtopography of polygonal tundra collect water from precipitation and snowmelt events, resulting in a
slow-release from the landscape through the thawed subsurface to surface water drainage networks (Helbig et al., 2013). This process has important implications for the leaching of dissolved organic carbon from active layer soils and for advective heat transfer through the subsurface. The role that polygonal tundra regions play in global terrestrial biogeochemical feedbacks as the Arctic warms necessitates a greater understanding of their hydrologic flow regimes.

Polygonal tundra forms in cold environments by the cyclic process of vertical cracking of frozen ground due to thermal
contraction, water infiltration into these cracks, freezing of this infiltrated water, and subsequent re-cracking. Over many cycles, this process leads to the growth of subsurface ice-wedges connected in polygonal patterns known as ice-wedge polygons (Liljedahl et al., 2016). Ice-wedge polygons vary in size from several meters to a few tens of meters in diameter and develop over time frames of hundreds to thousands of years (Leffingwell, 1915; Lachenbruch, 1962; Mackay, 2000; Abolt and Young, 2020). Thermal expansion of the ice and soil during summer months often results in warping the soil strata, forming parallel
rims on both sides of the ice-wedge and a trough directly above the ice-wedge (refer to Figure 1 and 2). Ice-wedge polygons with well-formed rims and troughs with a distinct, central topographic depression are referred to as low-centered polygons (as opposed to high-centered or transitional polygons). Ponding occurs frequently in the centers because the rims can trap water during snowmelt or precipitation events. This ponded water subsequently drains through the subsurface to the troughs (Helbig et al., 2013; Koch et al., 2018; Wales et al., 2020).

Recent observational studies have shaped our current conceptualization of low-centered polygonal tundra hydrology (Boike et al., 2008; Helbig et al., 2013; Koch et al., 2014; Liljedahl et al., 2016; Koch, 2016; Koch et al., 2018) indicating a system of inundated polygonal centers surrounded by elevated rims that prevent surface water from flowing overland into surrounding trough ponds. This land formation results in standing water over the polygon centers, reduces immediate landscape runoff from precipitation events, and increases evaporation (Liljedahl et al., 2016). Additionally, the elevated ponded water in polygon
centers produces hydrological gradients that result in sustained outward flow through the subsurface under the rims, a process

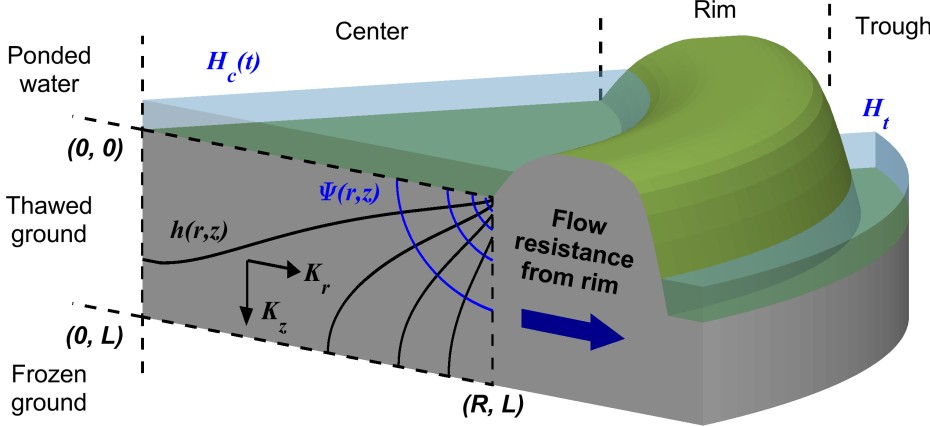

**Figure 1.** Pie-wedge schematic diagram of 3D-axisymmetric analytical model of inundated low-centered polygon drainage. The diagram represents an idealized *pie wedge* slice of a low-centered polygon including a wedge of the ponded center, rim, and trough. Equipotential hydraulic head lines are denoted as $h(r,z)$ and streamlines as $\Psi(r,z)$. $L$ is the depth of the thawed soil layer and $R$ is the radius of the polygon center. The ponded water height ($H_c(t)$) and trough water level ($H_t$) are noted.

that has been observed at several field sites (Helbig et al., 2013; Liljedahl and Wilson, 2016; Koch, 2016; Wales et al., 2020). This lateral flow controls landscape redistribution of water during the summer months (Helbig et al., 2013), governs ponded water budgets (Koch et al., 2014; Koch, 2016), and makes up a notable portion of regional river discharge (King et al., 2020). The manner and timing with which polygonal tundra landscapes transition from inundated to drained conditions has important
implications for (1) transitions from atmospheric emissions of methane to carbon dioxide (Conway and Steele, 1989; Moore and Dalva, 1997; Zona et al., 2011; Zhu et al., 2013; O'Shea et al., 2014; Throckmorton et al., 2015; Wainwright et al., 2015; Lara et al., 2015; Vaughn et al., 2016), (2) dissolved organic carbon emissions to surface waters (Zona et al., 2011; Abnizova et al., 2012; Laurion and Mladenov, 2013; Larouche et al., 2015; Plaza et al., 2019), (3) biological succession (Billings and Peterson, 1980; Jorgenson et al., 2015; Wolter et al., 2016), and (4) ground surface deformation (Mackay, 1990, 2000; Raynolds
et al., 2014; Nitzbon et al., 2019).

    The microtopographic features of polygonal tundra result in pronounced fine-scale spatial gradients in thermal and hydrologic conditions (Boike et al., 2008; Zona et al., 2011; Helbig et al., 2013) leading to sharply contrasting biogeochemical conditions (Newman et al., 2015; Norby et al., 2019). Standing water in polygonal centers shows distinctly different chemistry compared to surface water in troughs and ponds (Zona et al., 2011; Koch et al., 2014, 2018) which has been attributed to shallow
subsurface flow from centers to troughs (Koch, 2016). For example, Koch et al. (2014) observed high nutrient concentrations in troughs adjacent to centers with low concentrations, indicating drainage from the latter to the former. Given that the ice-wedge polygon can be considered the fundamental hydrologic landscape unit that initiates landscape-scale surface/subsurface flow and discharge, understanding their characteristic drainage pathways and residence times provides a basis to quantify the

transport of terrestrially sourced dissolved organic carbon to surface waters. This is important for carbon emissions because dissolved organic carbon in surface waters can be mineralized and released to the atmosphere (Raymond et al., 2013).

Although there have been numerous field observations of inundated low-centered polygon drainage to troughs (Helbig et al., 2013; Koch et al., 2014; Wales et al., 2020), the research here is, to our knowledge, the first in-depth investigation into the relative pathways and timing associated with inundated ice-wedge polygon drainage based on hydrologic first principles. Aside from the field tracer experiments of Wales et al. (2020), previous research has primarily focused on 1D vertical hydrothermal effects within ice-wedge polygons (Atchley et al., 2015; Harp et al., 2016; Atchley et al., 2016) or polygonal tundra surface hydrology (Boike et al., 2008; Jan et al., 2018a, b). Other researchers have investigated the hydrology of multiple polygons without investigating the drainage pathways within individual ice-wedge polygons (Cresto-Aleina et al., 2013; Nitzbon et al., 2019, 2020). While basic hydrology dictates that ice-wedge polygon geometry and heterogeneity will explicitly govern subsurface drainage pathways and duration, to date, a rigorous hydrological investigation of this process has not been presented.

The geometric shape of low-centered polygons along with soil hydraulic properties (for example, hydraulic conductivity) affects the distribution of hydraulic heads that control the pathways and timing of inundated ice-wedge polygon drainage. In this paper, we use a recently developed model (Zlotnik et al., 2020) based on hydrogeological first principles (Harr, 1962; Cedergren, 1968; Freeze and Cherry, 1979; Bear, 1979) to understand and gain intuition into the effects of geometry and hydraulic properties on inundated ice-wedge polygon drainage.

This model idealizes the thawed subsurface of an inundated low-centered polygon as a thin cylinder overlain with an initial height of ponded water that drains through the cylindrical thawed soil layer to the surrounding trough (outer vertical boundary; an annular ring defined by a line from $(R, 0)$ to $(R, L)$ in Figure 1). The depth $(L)$ and radius $(R)$ of the thawed soil layer of the polygon center and horizontal (radial) and vertical hydraulic conductivity are parameters of the solution. It is important to note that we are focusing on the drainage of inundated low-centered polygons here, and that biogeochemical investigations suggest that the drainage of non-inundated low-centered, transition, and high-centered polygons may have different characteristics (Heikoop et al., 2015; Newman et al., 2015; Wainwright et al., 2015; Wales et al., 2020).

We demonstrate that the model is able to accurately simulate ice-wedge polygon drainage by performing a calibration to field measurements over an entire thaw season. We calibrate the model parameters to fit water levels measured in the center of a low-centered polygon within the Barrow Environmental Observatory near Utqiaġvik, Alaska. We force the model with measured trough water levels, precipitation, thaw depth, and evapotranspiration. The goodness-of-fit of initial calibration efforts indicated that the model was unable to capture the polygon-center water level trends (i.e., the model was falsified). However, by considering (1) increased precipitation and (2) a thaw depth dependent (and therefore time-variable) vertical hydraulic conductivity, the model produces a good fit to the water levels. Model refinement (1) accounts for pond accumulation due to runoff from surrounding microtopographic highs (e.g., rims) while (2) accounts for the decreasing hydraulic conductivity of deeper tundra soil layers. The calibration indicates the minimum refinements necessary to a hydrology model to capture ice-wedge polygon drainage dynamics and indicates the important factors in this process. This calibration, based on transient boundary conditions, provides confidence in the physical meaningfulness of the steady-state snapshots presented in the sensitivity analysis.

We then use the model to investigate the pathways and timing of inundated polygon drainage through a sensitivity analysis of polygon geometry and anisotropy. The analysis of the model enhances our intuition into the pathways and timing of inundated ice-wedge polygon drainage for polygons with different geometries and degrees of anisotropy. Varying the geometry of the cylinder not only allows us to capture relative differences in drainage between different sized polygons (in other words, polygons with different radii) but also seasonal and inter-annual variations in polygon thaw-layer depths within a single polygon (in other words, as the thawed soil layer increases during a single thaw season or as the active layer increases from year to year). By allowing for different effective hydraulic conductivities between the horizontal and vertical directions, we can evaluate the effects of preferential flow directions on drainage. In addition to geologic layering, processes such as frost heave and horizontal ice lens thaw can result in preferential horizontal flow in cold climates (Mackay, 1981; Matsuoka and Moriwaki, 1992; Wales et al., 2020). It has also been observed that high hydraulic conductivity peat layers overlying lower hydraulic conductivity mineral soil results in horizontal watershed drainage (McDonnell et al., 1991; Brown et al., 1999; Quinton and Marsh, 1999). Helbig et al. (2013) found that in polygonal tundra, the peat layer quickly redistributed precipitation events across the subsurface topography from the rims to the centers and troughs. Vertical cracks, voids left from decayed roots, and animal burrows can result in preferential vertical flow. The cylindrical geometry and anisotropy are not intended to cover all potential variations in ice-wedge polygons, but rather provide base cases where those variations will cause deviations around the idealizations considered here.

The calibration to field data indicates that the model is able to capture ice-wedge polygon drainage dynamics. Our sensitivity analysis provides a new perspective on inundated polygon hydrogeology indicating the effects of polygon geometry and hydraulic conductivity anisotropy on drainage pathways and timing. Although the simplifications of the model may limit its applicability to some scenarios, they allow general intuitive insights to be drawn which would be obfuscated without them. The findings here provide a basis to quantify and understand deviations from our idealized scenarios.

## 2    Methods

### 2.1    Model overview

We use recently developed 3D-axisymmetric analytical solutions, derived and validated in Zlotnik et al. (2020) and described in Appendix A, of hydraulic heads and the stream function in the thawed soil layer below a polygon center (dashed rectangle in Figure 1) to investigate inundated low-centered polygon drainage pathways. While we perform the calibration to field data using the dimensional form of the solution for hydraulic head, we perform the sensitivity analyses using the non-dimensional form of the solution for hydraulic head and stream function (refer to equation A7 and A8). The non-dimensional hydraulic heads and stream function are independent of time, representing the relative pattern of hydraulic heads and stream function throughout the drainage process. The dimensional values at different times would indicate the change in the absolute magnitude of these patterns as the drainage process proceeds. To evaluate drainage pathways, we plot flow nets, as illustrated in Figure 1, composed of lines of equal non-dimensional hydraulic head ($h^*(r^*, z^*)$), referred to as equipotentials, and contours of the

stream function ($\Psi^*(r^*, z^*)$) as a function of radius and depth. The stream function contours are referred to as *streamlines* and are used here to identify steady drainage pathways.

Non-dimensional hydraulic head lines are drawn from 0.05 to 0.95 by increments of 0.1. Dimensional heads ($h(r, z, t)$) can be obtained from the non-dimensional heads ($h^*(r^*, z^*)$) using the ponded height of water in the polygon center $H_c(t)$ and the height of water in the trough $H_t$ as

$$h(r, z, t) = H_t + (H_c(t) - H_t)h^*(r^*, z^*), \; r^* = \frac{r}{L}\sqrt{\frac{K_z}{K_r}}, \; z^* = \frac{z}{L}, \tag{1}$$

where $r^*$ and $z^*$ are non-dimensional radius and depth, respectively, $L$ is the polygon thaw depth, and $K_r$ and $K_z$ are the radial (horizontal) and vertical hydraulic conductivity, respectively (refer to Figure 1). The aspect ratio is defined as $R/L$ and coefficient of anisotropy is $K_r/K_z$.

We use a Robin boundary condition (a third-type boundary condition allowing both head and flux to be specified) for the outer vertical boundary. This allows the model to represent the resistance of drainage under the rim due to elevated frozen ground and due to the accumulation of fine soil particles at the soil/water interface of the trough (Koch et al., 2018). The resistance to drainage below the polygon rim and across the soil/water interface into the trough is represented by a hydraulic conductance $\kappa = k/l$, where $k$ is the hydraulic conductivity and $l$ is the thickness of the drainage interface (refer to equation A3).

We normalize the stream function $\Psi(r^*, z^*)$ from 0 to 1, and similar to heads, plot stream function contours (streamlines) from 0.05 to 0.95 by increments of 0.1. The region between any two adjacent streamlines conveys 10% of the drainage, while the two remaining regions (less than 0.05 and greater than 0.95), convey 5% of the drainage each. To make quantitative comparisons of the relative spread of drainage between polygons, we calculate the percent of each polygon volume accessed by 95% of the drainage (polygon volume with non-dimensional stream function > 0.05).

The change in non-dimensional ponded water height in the polygon center due solely to drainage (the depletion curve) over time can be expressed by a simple exponential decay function as

$$H_c^*(t) = e^{-t/t_L} \tag{2}$$

where $t_L$ is the characteristic time of drainage (refer to equation A11), defined as the time when the ponded height is $1/e$ times, or $\sim 37\%$ of, its original height $H_{c,0}$. The dimensional ponded height $H_c(t)$ can be obtained as

$$H_c(t) = H_t + (H_{c,0} - H_t)H_c^*(t). \tag{3}$$

As described in Zlotnik et al. (2020), for the calibration, temporally changing trough water levels, precipitation, evapotranspiration, and thaw depth are accounted for by running the model in a piecewise fashion.

## 2.2 Calibration approach

We performed nonlinear least-squares calibrations using a Levenberg-Marquardt approach (Transtrum et al., 2011). The objective function is the sum-of-squared errors (SSE) between the measured and simulated polygon-center water levels expressed

as

$$\text{SSE}(\boldsymbol{\theta}) = \sum_{i=0}^{N} (H_c(t_i, \boldsymbol{\theta}) - H_c^m(t_i))^2, \tag{4}$$

where $\boldsymbol{\theta}$ is a vector of model parameters and $H_c^m(t_i)$ is the measured polygon-center water level at the $i$th time. We present three calibration cases including (1) a base case (model setup similar to the validation in Zlotnik et al. (2020)), (2) precipitation multiplied by a calibrated factor, and (3) a depth-dependent vertical hydraulic conductivity ($K_z = f(D)$). Depending on the calibration case, $\boldsymbol{\theta}$ contains different parameters.

In calibration case 1, $\boldsymbol{\theta}$ includes parameters $K_r$, $K_z$, $\kappa$, and $H_{c,0}$. Based on field observations and the validation in Zlotnik et al. (2020), we constrain $K_z$ to be less than $K_r$ using a meta-parameter $\mathcal{F}_{K_z}$ as $K_z = K_r \sigma(\mathcal{F}_{K_z})$ where $\sigma$ is the sigmoid function defined as $\sigma(x) = 1/(1 + \exp(x))$.

In calibration case 2, we add a precipitation multiplier $M_P$ to case 1, where $\hat{P} = M_P P$ and $\hat{P}$ is the augmented precipitation accounting for runoff from microtopographic highs and potential rain gauge under catch.

In calibration case 3, we add vertical hydraulic conductivity as a super-elliptical function of thaw depth to calibration case 2 as $K_z(D) = K_z^{min} + (K_z^{max} - K_z^{min})(1 - D_{norm}^a)^{1/a}$, $D_{norm} = \frac{D - D^{min}}{D^{max} - D^{min}}$, where $K_z^{max}$ and $K_z^{min}$ are the maximum and minimum values of $K_z$, respectively, $D^{max}$ and $D^{min}$ are the maximum and minimum values of $D$ during the thaw season, respectively, and $a$ is the super-elliptical shape parameter that controls curvature. $K_z^{max}$ is constrained to be positive using a meta-parameter $\mathcal{F}_{K_z^{max}}$ and the softplus function as $K_z^{max} = log(1 + \exp(\mathcal{F}_{K_z^{max}}))$. We ensure that $K_z^{min} \leq K_z^{max}$ using a meta-parameter $\mathcal{F}_{K_z^{min}}$ as $K_z^{min} = K_z^{max} \sigma(\mathcal{F}_{K_z^{min}})$. We ensure that the shape parameter $a$ is restricted between 0.5 and 1.5 using a meta-parameter $\mathcal{F}_a$ as $a = 0.5 + 1.5\sigma(\mathcal{F}_a)$. The constrained super-elliptical function allows the calibration to identify a general nonlinear decrease in vertical hydraulic conductivity with increasing thaw depth.

## 2.3 Acquisition of field data used in calibration

The measured data used in the calibration were collected during the thaw season of 2013 from June 16 to September 18 from a low-centered polygon within the Barrow Environmental Observatory near Utqiaġvik, Alaska (Figure 2). We used water levels collected by Liljedahl and Wilson (2016) with 0.2 cm resolution and precipitation collected by Hinzman et al. (2014) with a 0.1 mm resolution. We determined thaw depths from thermal transects collected by Romanovsky et al. (2017) with an accuracy of $0.1^\circ$ C. Due to a lack of continuous local evapotranspiration measurements, we obtained evapotranspiration data from NASA's Global Land Data Assimilation System (GLDAS) (Rodell et al., 2004).Given the continuous (albeit diurnally fluctuating), low magnitude evapotranspiration signal, its effect on our calibration is relatively insignificant compared to the sporadic precipitation events that drive large scale fluctuations in water levels. Therefore, in lieu of local evapotranspiration measurements, the GLDAS evapotranspiration is deemed sufficient for our purposes here.

## 2.4 Parameter ranges for sensitivity analyses

We selected aspect ratio and anisotropy scenarios based on existing literature and observations. While a pan-Arctic survey of ice-wedge polygon diameters does not to our knowledge currently exist, researchers have provided general characteristics

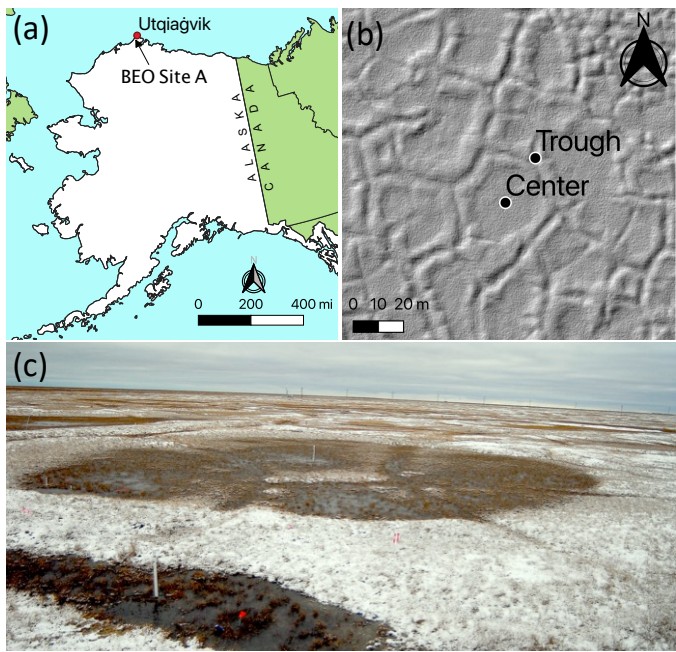

**Figure 2.** (a) Map, (b) hillshade relief with well locations, and (c) photo of the Barrow Environmental Observatory (BEO) Site A near Utqiaġvik, Alaska. Photo courtesy of Bob Busey.

based on extensive observations. Leffingwell (1915) states that ice-wedge polygons have an estimated average diameter of around 15 m. Liljedahl et al. (2016) indicate that low-centered polygons (including rims) can have diameters from 5-30 m. Abolt et al. (2018) state that polygonal formations in ice-wedge tundra are 10-30 m in diameter. Based on digital elevation

models, Abolt et al. (2019) calculate that the mean polygon diameter near Utqiaġvik, Alaska is around 15 m with 5th and 95th percentiles of approximately 8 and 33 m, respectively. Using a similar analysis on data from Prudhoe Bay (Abolt and Young, 2020), the mean polygon diameter is around 13 m with 5th and 95th percentiles of approximately 7 and 30 m, respectively.

Maximum thaw depths are increasing throughout the Arctic (Jorgenson et al., 2006). Deeper thaw depths may ultimately result in low-centered polygon transition to high-centered polygons, at which time polygon-center inundation will cease to

210 occur due to rim collapse. Therefore, there is a practical limit to the thaw depth of interest in our analysis. Lewkowicz (1994) reported that active layer thickness varied from 40 to 90 cm within the polygonal tundra of Fosheim Peninsula, Ellesmere Island, Canada in 1994. Shiklomanov et al. (2010) reported that polygonal tundra at several study sites near Utqiaġvik, Alaska monitored from 1995 to 2009 had active layer thicknesses from 17 to 47 cm. Based on extensive ground-penetrating radar surveys near Utqiaġvik, Alaska in 2013, Jafarov et al. (2017) document that average active layer thickness was 41 cm with a

215 standard deviation of 9 cm. Even for polygons with substantial active layer thickness, it is important to understand the change in drainage throughout the season by evaluating early-season thaw depths, which are captured in our analysis by larger aspect (i.e., radius to thaw depth) ratios.

Based on these considerations, we evaluate aspect ratios from 2.5 to 20. For example, given a thaw depth of 1 m, an aspect ratio of 2.5 would represent a 2.5 m radius polygon center, while an aspect ratio of 20 would represent a 20 m radius polygon center, more than covering the observed range. To consider cases with thinner thawed soil layers, larger aspect ratios can be used. For example, given a thaw depth of 0.5 m, an aspect ratio of 20 would represent a polygon with a 10 m radius center.

Comprehensive measurements of hydraulic conductivity anisotropy are lacking from ice-wedge polygons. Based on tracer arrival times, Wales et al. (2020) estimated horizontal conductivities of approximately 0.7 to 84 m $d^{-1}$ for a low-centered polygon and 0.01 to 0.3 m $d^{-1}$ for a high-centered polygon. Wales et al. (2020) stated that these are considered lower bound estimates of horizontal conductivity since, based solely on breakthrough times, the approach is unable to isolate horizontal and vertical flow effects on arrival times. Beckwith et al. (2003) performed laboratory measurements of anisotropy on small samples from peat soils where horizontal conductivities were around 6-7 m $d^{-1}$ and vertical conductivities were around 0.2-0.4 m $d^{-1}$ indicating preferential horizontal flow. The soil samples used in Beckwith et al. (2003) were from peat bogs in England, and therefore would not have been subjected to similar freeze-thaw dynamics as contemporary polygonal tundra soils, which would presumably lead to even greater preferential horizontal flow. Therefore, these estimates may provide a lower bound estimate for Arctic peat horizontal hydraulic conductivity. Based on the existing literature and above considerations, we considered hydraulic conductivities from 0.005-5 m $d^{-1}$, primarily focusing on anisotropic values from 0.1 to 100. These values allow us to investigate the relative effects of anisotropy on the drainage pathways and timing using hydraulic conductivities consistent with currently available measurements.

A physical interpretation of our selected values of anisotropy coefficient can be obtained by considering that ice-wedge polygon soils are typically layered and that the horizontal and vertical hydraulic conductivities can therefore be considered as *effective* properties. As such, the effective horizontal hydraulic conductivity captures parallel flow dominated by the higher conductivity layers and the effective vertical hydraulic conductivity captures flow in series across multiple layers and is dominated by the lower conductivity layers. Therefore, an anisotropy coefficient of 100 would *effectively* capture layers with 2 orders of magnitude difference in hydraulic conductivity, while an anisotropy coefficient of 0.1 would capture the hypothetical scenario where vertical cracks or burrows result in preferential vertical flow. Given the current lack of direct measurements of ice-wedge polygon anisotropy coefficient, we cover a broad range of possible scenarios.

The hydraulic conductivity of the interface between soil and open water can be half that of the rest of the soil due to the accumulation of fines (Koch et al., 2018). The hydraulic conductance under the rim may also impede drainage due to a raised permafrost table following the surface topography. Note that while the raised permafrost table under the rim will constrict flow and alter drainage pathways, hydrologic first principles indicate that this effect will be restricted to the region of the model near the rim. This is analogous to the effect of a partially penetrating well on flow in an aquifer, which dissipates quickly and is non-existent by a lateral distance of 1.5 to 2 times the aquifer thickness for isotropic aquifers (Bear, 1979). This effect will diminish with increasing aspect ratio and decreasing anisotropy coefficient, and will not alter the qualitative insights drawn from the relative comparison of drainage pathways in our analysis. It is conceivable that early in the thaw season the permafrost table under the rim could extend above the center ground surface within the vertical interval of the center pond. In this case, there will likely be very little drainage occurring associated with a very small discharge conductance. Although the conductance of

the outer interface (defined as $\kappa$ above) can therefore influence drainage, a preliminary analysis indicated that in most cases, its effect on drainage was significantly less than that of aspect ratio and anisotropy coefficient (Zlotnik et al., 2020). Therefore, we chose to use a default value for $\kappa$ in all cases (unless otherwise specified) of 5 d$^{-1}$.

In practice, as in our calibration, the water level in troughs ($H_t$ in equations 1 and 3) will vary throughout the thaw season, affecting the magnitude of heads in the soil of the polygon center and drainage times. As the non-dimensional heads are relative to $H_t$ (refer to equation A7), its value does not affect our relative comparisons of drainage patterns (which are based on non-dimensional heads that are normalized from 0 to 1). The value of $H_t$ will affect our comparisons of drainage time, but in a systematic, interpretable manner. For example, a higher $H_t$ will compress the exponential curve defined by equation 3 upwards, while a lower $H_t$ will stretch the exponential curve downwards. In cases where $H_t$ is below the polygon-center ground surface, the solution is valid until $H_c$ reaches the ground surface, at which time the ponded center has completely drained. Therefore, to isolate our analysis to aspect ratio and anisotropy, we have set $H_t$ in all cases equal to the polygon-center ground surface.

Although included in the calibration, we have neglected the effects of evaporation and precipitation in the sensitivity analysis as they will not affect the drainage patterns we present (based on non-dimensional heads) and their effect on drainage timing (based on non-dimensional depletion curves) is straightforward, shifting the non-dimensional exponential drainage curve upwards or downwards. In other words, using non-dimensional variables is a powerful approach to gain intuition into the fundamentals of inundated ice-wedge polygon drainage irrespective of variable magnitude.

## 3 Results

We verify the model through calibration to water-level measurements, identifying refinements necessary for hydrologic models to match field observations of polygon drainage. We present drainage flow nets for various aspect ratios and anisotropy coefficient values, polygon-center ponded height depletion curves, and maps of the percent of the thawed soil accessed by 95% of the drainage flow and depletion characteristic times as a function of aspect ratio and anisotropy coefficient. We compare the effect of aspect ratio on drainage pathways in an isotropic and a highly anisotropic ($K_r/K_z = 100$) polygon. The drainage pathways are also evaluated for various anisotropies holding the aspect ratio constant. We present a global perspective of the combined effects of aspect ratio and anisotropy coefficient on the focusing/spreading of the drainage flow by mapping the percent of the polygon thawed soil volume that is accessed by 95% of the drainage flow. We evaluate the effect of aspect ratio and anisotropy coefficient on drainage time by plotting the polygon-ponded water height depletion curves. Similar to accessed volume, we provide a global perspective on the combined effects of aspect ratio and anisotropy coefficient on drainage time by mapping the depletion characteristic time. We illustrate the counteracting effects of aspect ratio and anisotropy coefficient on drainage pathways by showing two, although geometrically and hydrologically dissimilar, mathematically equivalent solutions of drainage.

## 3.1 Calibration to field measurements

We present the best-fit polygon-center water levels for the calibration cases along with the measured values in Figure 3. The two dominant drivers of the polygon-center water levels, the measured trough water level and precipitation, are shown for reference. Calibration case 1 (green line) is unable to match the high and low points of the water level measurements, and instead follows a more medial path over time (RMSE=1.42 cm; $R^2$=0.49; refer to Table 1). This indicates that the measured precipitation alone is not able to account for the increased water levels after rain events. We also consider the calibration as failed (the model as falsified (Popper, 2005)) because the calibrated value of the discharge conductance ($\kappa = 835.3$ d$^{-1}$; refer to Table 1) is extremely large and non-physical.

Considering that precipitation will runoff from rims and collect in the polygon-center pond and that rain gauges may have under catch issues (e.g., (Pollock et al., 2018)), we performed calibration case 2, also shown in Figure 3 (purple line), which includes a precipitation multiplier. Calibration case 2 significantly improves the fit (RMSE=0.89 cm; $R^2$=0.80), but still has significant mismatch as the limitations of the model require the calibration to compromise between matching water levels at early and late times. As in calibration case 1, we also consider the model for calibration case 2 as falsified because the calibrated value of $\kappa = 1529.2$ d$^{-1}$ is even larger than in case 1, and therefore even less physical.

The compromised fit in calibration case 2 indicated that the effective vertical hydraulic conductivity likely decreases as the thaw depth increases, which would be consistent with observations of reduced hydraulic conductivities at depth. In calibration case 3, we implemented vertical hydraulic conductivity as a function of thaw depth. This model refinement allowed the calibration to achieve a good overall fit to the data (RMSE=0.37 cm; $R^2$=0.96) as shown in Figure 3 (red line). $K_r$ is well within the estimated limits of 0.7 to 84 m d$^{-1}$ based on a tracer test at a nearby low-centered polygon conducted by Wales et al. (2020). Based on a rim width of 1 meter, the discharge conductance of $\kappa = 3.3$ d$^{-1}$ would correspond to a drainage boundary layer hydraulic conductivity of 3.3 m d$^{-1}$, which is also physically reasonable as opposed to the first two calibration estimates.

The standard errors of the calibrated parameters for calibration case 3 listed in Table 1 indicate how well constrained the parameters are by the calibration. It is apparent that the hydraulic conductivities (horizontal and minimum and maximum vertical) are not well constrained with relatively large standard errors. These parameters (or their meta-parameters) also have large covariances with each other indicating their correlated effect on the model. However, despite the lack of constraint of these parameters due to their correlated effect on the model, the calibration does identify reasonable values for them. The standard errors of the discharge conductance, initial polygon-center water level, and precipitation multiplier indicates that they are well constrained by the calibration.

The calibration verifies that the model is able to capture ice-wedge polygon drainage characteristics. In the next sections, we perform sensitivity analyses using non-dimensional forms of this verified analytical solution to gain insights into ice-wedge polygon drainage characteristics. The use of non-dimensional solution snapshots eliminates the need to consider the precipitation multiplier and thaw-depth dependent vertical hydraulic conductivity explicitly. Instead, their effects are implicit in the relative differences between snapshots.

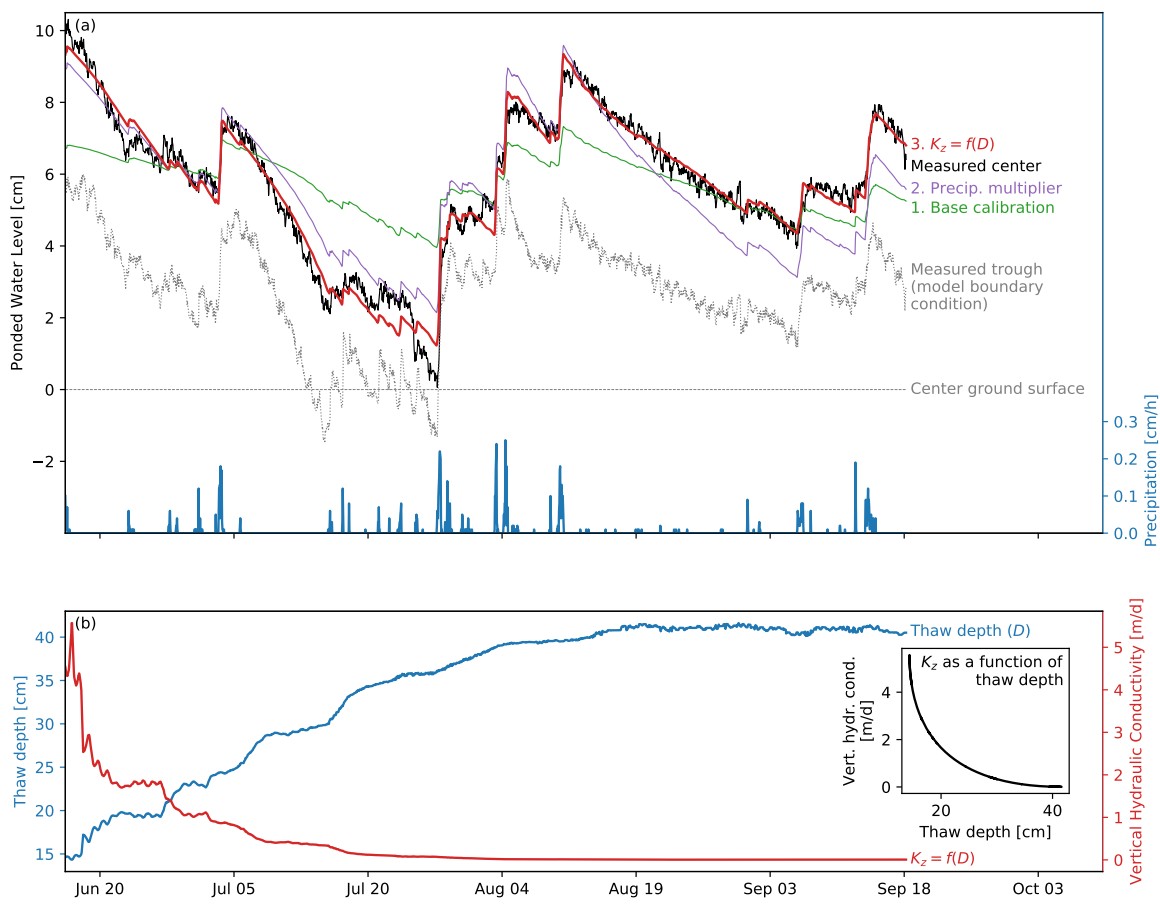

**Figure 3.** (a) Progression of center water-level calibration. The (1) base calibration, (2) calibration with a precipitation multiplier, and (3) calibration with vertical hydraulic conductivity as a function of thaw depth ($K_z = f(D)$). The measured polygon-center and trough water levels are plotted for reference. Precipitation is plotted on the right y-axis. (b) Measured thaw depth and calibrated vertical hydraulic conductivity as a function of thaw depth.

## 3.2 Drainage pathways

The relative effect of aspect ratio on drainage pathways when the hydraulic conductivities are isotropic ($K_r/K_z = 1$) is shown in Figure 4. Under isotropic hydraulic conductivities, results show that the drainage pathway for high aspect ratio polygons will be predominantly isolated to an annular region at the periphery of the polygon center. As the aspect ratio decreases (in

**Table 1.** Calibrated parameters values, root-mean-squared error (RMSE), coefficient of determination ($R^2$) for calibration cases, and parameter standard errors for case (3). Note that the coefficient of determination is equivalent to the Nash-Sutcliffe Efficiency here. Dashes indicate that the parameter was not part of the calibration.

| Parameter | Symbol | Calibration Case (1) Base | (2) Precip. mult. | (3) $K_z = f(D)$ | Standard error |
|---|---|---|---|---|---|
| Hor. Hydr. Cond. [m d$^{-1}$] | $K_r$ | 9.8 | 36.7 | 19.9 | 52.3 |
| Vert. Hydr. Cond. [m d$^{-1}$] | $K_z$ | 1.02e-04 | 5.14e-04 | – | – |
| Discharge conductance [d$^{-1}$] | $\kappa$ | 835.3 | 1529.2 | 3.3 | 0.17 |
| Initial polygon-center water level [cm] | $H_c(t=0)$ | 6.58 | 8.66 | 9.12 | 4.25e-04 |
| Precipitation multiplier [–] | $M_P$ | – | 2.24 | 2.21 | 1.57e-2 |
| Min. Vert. Hydr. Cond. [m d$^{-1}$] | $K_z^{min}$ | – | – | 4.37e-03 | 28.4 |
| Max. Vert. Hydr. Cond. [m d$^{-1}$] | $K_z^{max}$ | – | – | 5.57 | 28.8 |
| Super-elliptical shape parameter [–] | $a$ | – | – | 0.50 | 2.0 |
| Root-mean-squared error [cm] | RMSE | 1.42 | 0.89 | 0.37 | – |
| Coefficient of determination | $R^2$ | 0.49 | 0.80 | 0.96 | – |

other words, as the thawed region of the polygon subsurface becomes deeper with respect to its width), the portion of the polygon accessed by drainage increases. For an aspect ratio of 20 (Figure 4a), 95% of the drainage is focused within around 6% of the polygon volume, while for an aspect ratio of 2.5 (Figure 4d), the drainage is spread over around 33% of the polygon volume. The spreading (increase in the accessed volume) occurs along the radial direction, where the horizontal extent of the accessed volume moves towards the middle of the polygon center ($r = 0$), while the vertical extent is nearly unchanged. The

results in Figure 4 indicate that throughout the thaw season as the thaw depth increases, or over successive years as the active layer thickens, the drainage path will spread out towards the middle of the polygon center. Similarly, Figure 4 can be used to evaluate drainage pathways of polygons of different widths but similar thaw depths. In this context, Figure 4 indicates that wider polygons will have more focused drainage, while drainage for smaller polygons will be more dispersed.

     The effect of anisotropy coefficient on drainage pathways when the aspect ratio is held constant ($R/L = 10$) is shown in

Figure 5. As the anisotropy coefficient increases (i.e., greater preferential horizontal flow), the region accessed by drainage flow becomes larger. Similar to a decreasing aspect ratio (smaller and/or more deeply thawed polygons) in Figure 4, increasing the anisotropy coefficient leads to a larger radial extent of the accessed region, while the vertical extent is nearly unaffected. When the vertical conductivity is ten times the horizontal (Figure 5a), only around 3% of the polygon volume is accessed by 95% of the drainage. If horizontal conductivity is 100 times vertical conductivity (Figure 5d), around 49% is accessed. These

results indicate that anisotropy has a significant impact on ice-wedge polygon drainage pathways.

     The effect of aspect ratio on drainage pathways when the hydraulic conductivities are highly anisotropic ($K_r/K_z = 100$) is shown in Figure 6. In this case, contrary to the isotropic case in Figure 4, as the aspect ratio decreases, the accessed volume generally decreases. However, the largest accessed volume is for an aspect ratio of 10 (Figure 6b), not 20 (Figure 6a). This

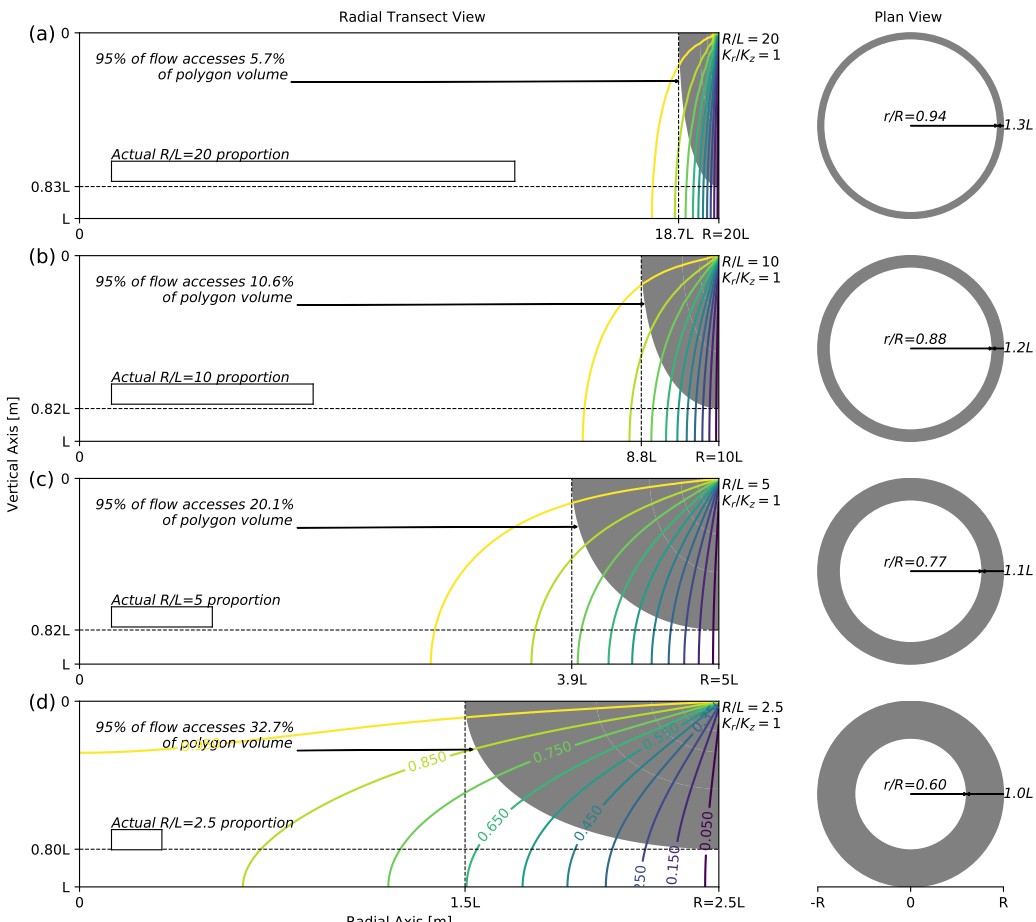

**Figure 4.** Effect of polygon aspect ratio on polygon drainage with isotropic hydraulic conductivity. Plots along the left contain polygon radial transect head contours (colored lines) and filled stream tubes (grey regions) for several polygon aspect ratios (radius/thickness). The grey shaded region denotes the portion of the transect accessed by 95% of the flow. The plots along the right contain corresponding grey rings indicating the surface area where 95% of the polygon flow infiltrates. Each plot along the left contains a rectangle drawn to the actual proportions for the given polygon aspect ratio. In all cases, the anisotropy coefficient (horizontal/vertical conductivity) is fixed at unity.

nuance in the dependence of accessed volume to aspect ratio with high anisotropy coefficient is due to the competing effects of radial extension and vertical contraction of the accessed volume as aspect ratio decreases. By comparing Figures 4 and 6, it is also apparent that the volume accessed by drainage is generally larger with higher anisotropy coefficient.

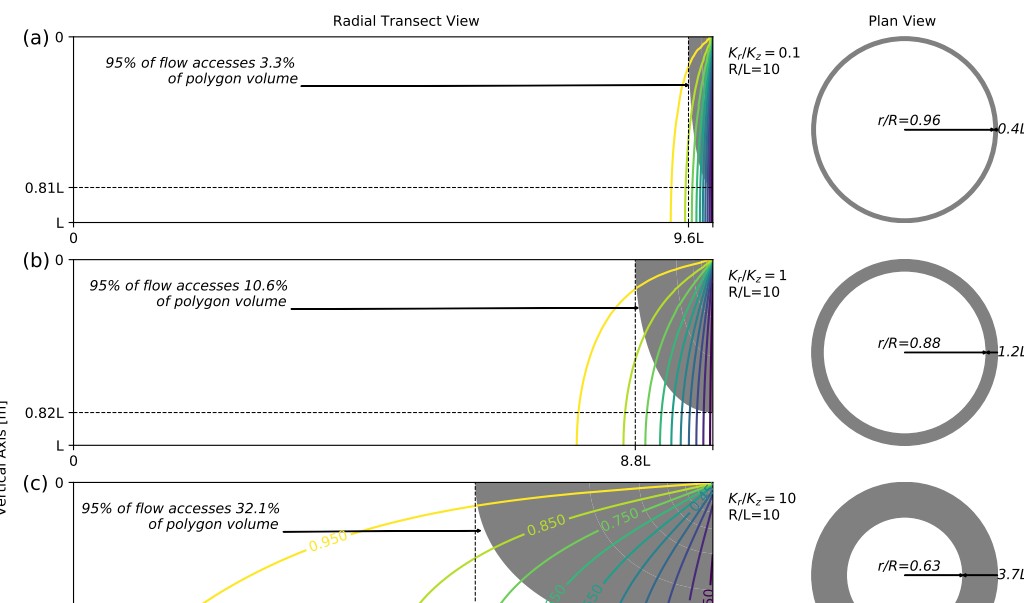

**Figure 5.** Effect of hydraulic conductivity anisotropy coefficient on polygon drainage. Plots along the left contain polygon radial transect head contours (colored lines) and filled stream tubes (grey regions) for several anisotropies (horizontal/vertical conductivity). The grey shaded region denotes the portion of the transect accessed by 95% of the flow. The plots along the right contain corresponding grey rings indicating the surface area where 95% of the polygon flow infiltrates. Each plot along the left contains a rectangle drawn to the actual proportions for the given polygon aspect ratio. In all cases, the polygon aspect ratio (radius/thickness) is fixed at 10.

To gain a global perspective on the trends in drainage pathways with aspect ratio and anisotropy coefficient, Figure 7 maps the percent of the polygon accessed by 95% of the drainage as a function of aspect ratio and anisotropy coefficient. This illustrates the overall structure of the combined effect of aspect ratio and anisotropy coefficient on accessed volume (focused

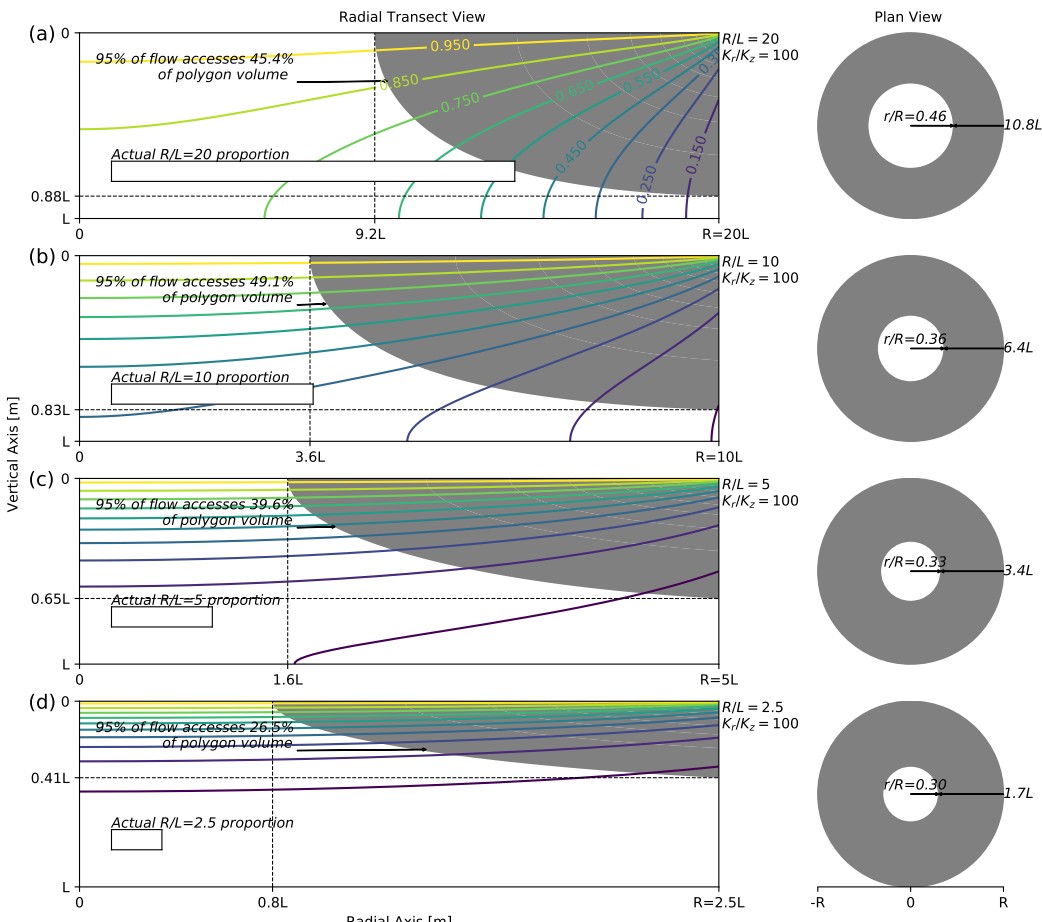

**Figure 6.** Effect of polygon aspect ratio on polygon drainage with high hydraulic conductivity anisotropy coefficient ($K_r/K_z = 100$). Plots along the left contain polygon radial transect head contours (colored lines) and filled stream tubes (grey regions) for several polygon aspect ratios (radius/thickness). The grey shaded region denotes the portion of the transect accessed by 95% of the flow. The plots along the right contain corresponding grey rings indicating the surface area where 95% of the polygon flow infiltrates. Each plot along the left contains a rectangle drawn to the actual proportions for the given polygon aspect ratio. In all cases, the anisotropy coefficient (horizontal conductivity/vertical conductivity) is fixed at 100.

versus dispersed drainage). As indicated by the annotated points, the trend in accessed volume with respect to aspect ratio in Figure 4 is represented along the anisotropy coefficient=1 transect, while the trend in Figure 6 is represented by the anisotropy

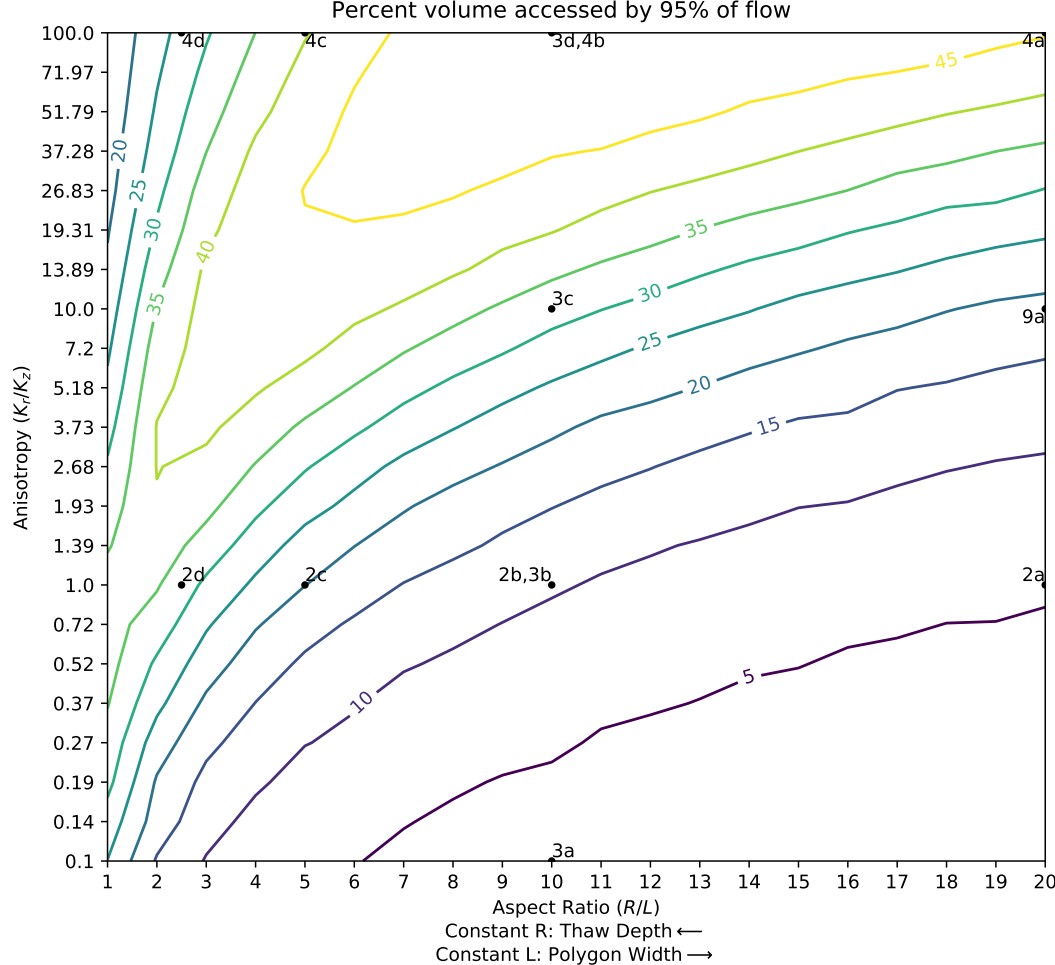

**Figure 7.** Contour map of the percent of the polygon access by 95% of the flow as a function of polygon aspect ratio and anisotropy coefficient. Locations associated with other figures are indicated by labeled points. By choosing a constant value for $R$ $(L = R/x)$, horizontal transects parallel to the x-axis represent the effect of thaw depth increasing from right to left, while they represent increasing polygon width from left to right for constant values of $L$ $(R = Lx)$. Note that $\kappa = 5\ \mathrm{day}^{-1}$ in these calculations, and therefore, the location associated with Figure 11b cannot be indicated.

coefficient=100 transect. The trend in accessed volume with respect to anisotropy coefficient in Figure 5 along the aspect ratio=10 transect is likewise indicated. There is a curved, spreading region of high accessed volume with increasing aspect ratio and anisotropy coefficient apparent in the contours of Figure 7. This curved feature represents the optimal balance of radial and vertical extension of the accessed volume region, as described in the previous paragraph. This structure explains the

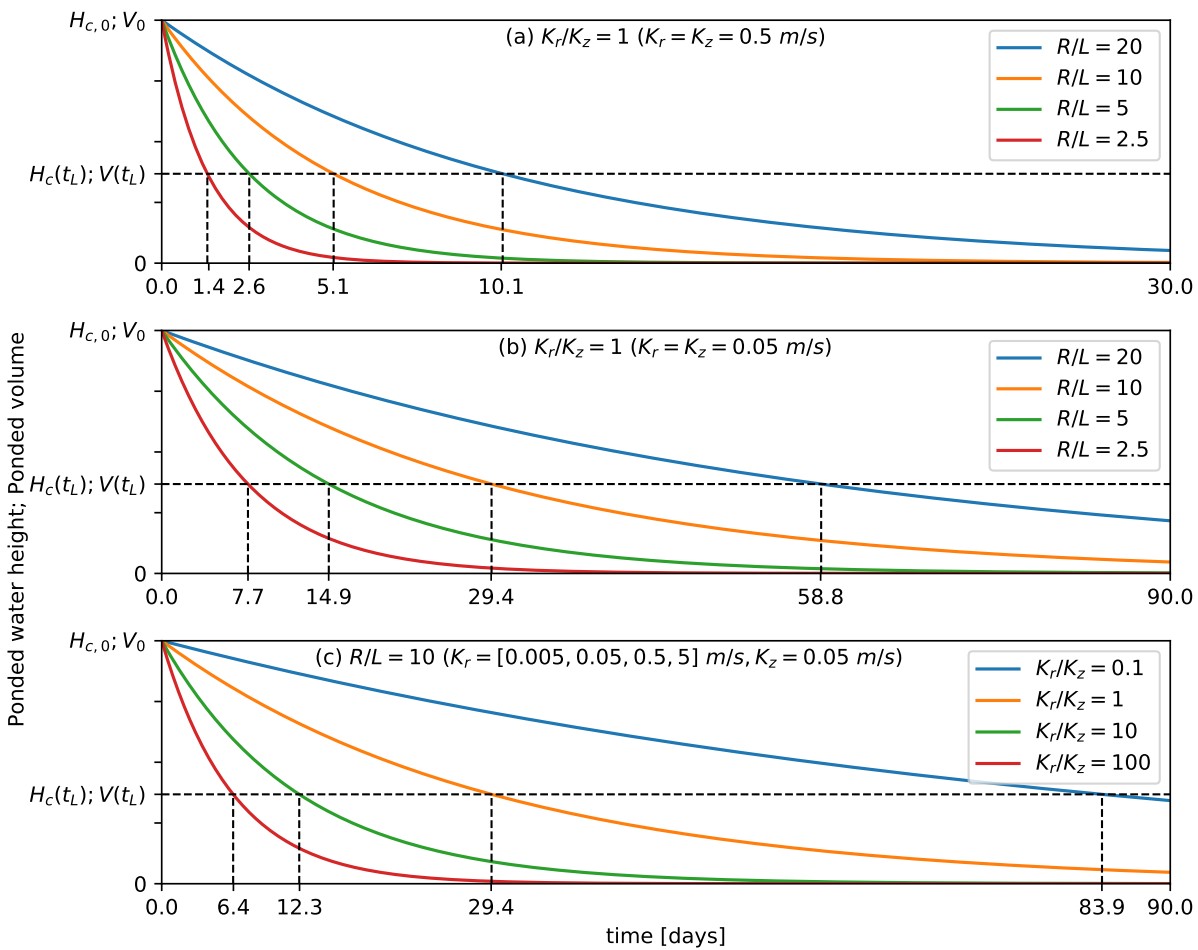

**Figure 8.** Depletion of ponded water height over time ($H_c(t)$) for (a,b) alternative polygon aspect ratios (radius/depth ($R/L$)) and (c) anisotropy coefficient ($K_r/K_z$). Plots (a) and (b) have anisotropy coefficient fixed at unity with plot (a) having high conductivity ($K_r = K_z = 0.5$ m/s) and plot (b) low conductivity ($K_r = K_z = 0.05$ m/s). The aspect ratio is fixed at 10 in plot (c). The curves also describe the depletion of ponded water volume over time ($V(t)$). Dashed lines indicate the characteristic times ($t_L$) in each case. Note that the orange line in (b) and (c) are identical, $R/L = 10$ and $K_r/K_z = 1$.

increasing accessed volume with decreasing aspect ratio in Figure 4 and the maximum accessed volume at an aspect ratio of 10 in Figure 6. The drainage flow is most spread out (largest accessed volume) when the aspect ratio is large and the anisotropy coefficient is high (upper right of Figure 7). The drainage is the most focused (least accessed volume) when the aspect ratio is large and the anisotropy coefficient is low (lower right of Figure 7).

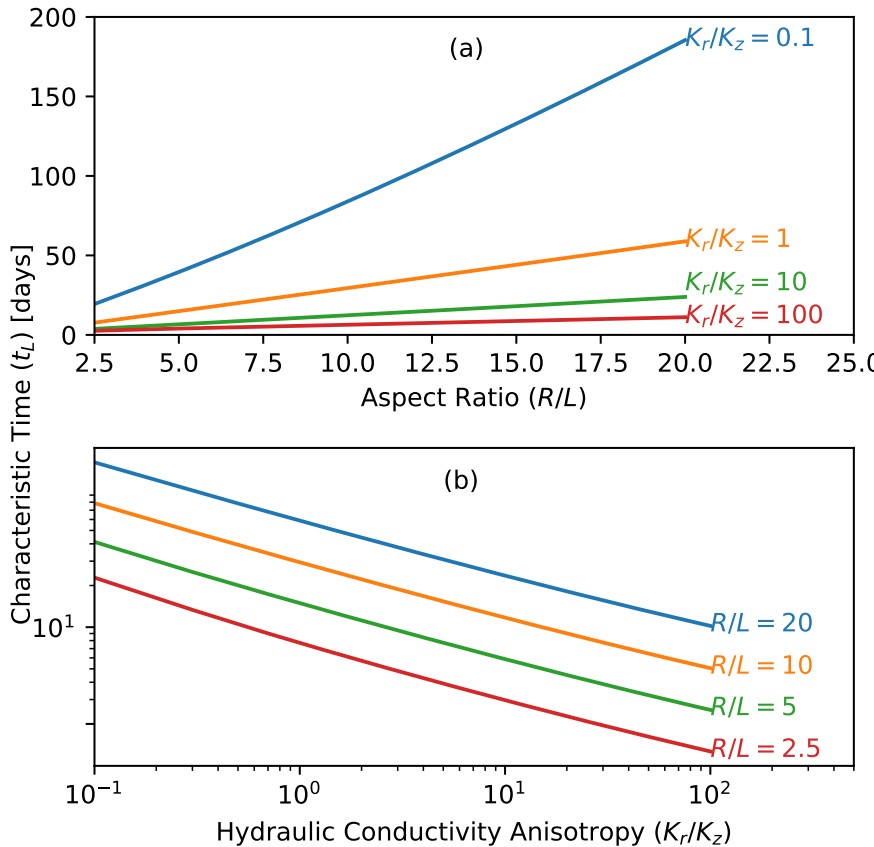

**Figure 9.** Characteristic time as a function of (a) polygon aspect ratio and (b) anisotropy coefficient. The $K_r/K_z = 1$ line in plot (a) corresponds to characteristic times in Figure 8b. Note that the trends in (a) are nearly linear and on natural scaled axes while the trends in (b) are also nearly linear on log transformed axes.

### 3.3 Ponded water depletion

Depletion curves for the non-dimensional ponded water height in the polygon center for various aspect ratios (at fixed high and low anisotropy coefficient) and anisotropies are shown in Figures 8a, b, and c, respectively. Note that since the depletion of the volume of ponded water is directly related to the ponded water height, the plots in Figure 8 can be used to obtain either, as indicated by the y-axis labels. As a point of reference between depletion curves, the characteristic time (the time when the height or volume of ponded water reaches $1/e \approx 0.37$ its initial height or volume) is indicated.

It is apparent from these plots that small, deeply thawed (lower aspect ratio) polygons will drain faster than wide, shallowly thawed (high aspect ratio) polygons, while polygons with higher anisotropy coefficient will drain faster than those with lower anisotropy coefficient. It is also apparent that the increase in drainage time with aspect ratio is nearly linear, while for anisotropy

coefficient, it is exponential. This is further illustrated in Figure 9, where we plot the trend in characteristic times as a function of aspect ratio for various anisotropies (Figure 9a) and as a function of anisotropies for various aspect ratios (Figure 9b). Drainage time (represented by characteristic time) increases in a nearly linear fashion, particularly for high anisotropies, with slight upward curvature increasing for low anisotropies. The drainage time decreases in a nearly log-log fashion (exponentially) with increasing anisotropy coefficient with an identical trend for different aspect ratios.

A global perspective on drainage timing trends, where characteristic time is mapped as a function of aspect ratio and anisotropy coefficient, is shown in Figure 10. The fastest (shortest) drainage times are achieved with a low aspect ratio and a high anisotropy coefficient, while the slowest (longest) drainage times are achieved with a high aspect ratio and low anisotropy coefficient. These trends indicate that given the same thawed soil layer thickness, wider polygons will drain more slowly than small polygons. Or, for polygons with similar aspect ratios, those with preferential horizontal flow will drain most quickly, while those with preferential vertical flow will drain most slowly.

## 3.4 Counteracting effects of aspect ratio and anisotropy

It is important to note that aspect ratio and anisotropy have similar effects on drainage. Within the model used here, in a similar fashion to aspect ratio, the anisotropy coefficient stretches the domain by multiplying the non-dimensional radius by $\sqrt{(K_z/K_r)}$ (refer to the definition of $r^*$ in equation 1). As a result, the effect of increasing anisotropy coefficient in the analytical solution is mathematically equivalent to a decrease in aspect ratio. In the end, the equipotential heads and streamlines computed on stretched or compressed radial coordinates ($r^*$) are assigned back to non-modified radial coordinates ($r$). This would lead one to believe that it should be possible to obtain two scenarios with different aspect ratios and anisotropies that produce the same mathematical solution. For example, doubling the aspect ratio should produce the mathematically identical effect as dividing the anisotropy coefficient by four. However, this is not the case because the solution involves a Biot parameter (Bi) which defines the ratio of the ability for fluid to conduct across the drainage interface (the vertical outer boundary) relative to the internal hydraulic conductivity as

$$\mathrm{Bi} = \frac{\kappa L}{\sqrt{K_r K_z}}, \tag{5}$$

where $\kappa$ characterizes the hydraulic conductance across the outer vertical boundary of the model (refer to Appendix A for more details). A large Bi indicates that the outlet boundary interface of the model will not limit the drainage, while a small Bi will result in outlet boundary interface limited drainage. Therefore, to obtain an identical mathematical drainage solution using different combinations of aspect ratio and anisotropy coefficient, one would also need to ensure that Bi is not modified in the process. However, given two solutions, this requires satisfying the conflicting conditions that $K_{z1}/K_{r1} = K_{z2}/K_{r2}$ and $K_{r1}K_{z1} = K_{r2}K_{z2}$ (refer to equations 1 and 5), where subscripts 1 and 2 refer to the two solutions. Since these two constraints cannot be simultaneously met, $\kappa$ or $L$ must also be modified along with aspect ratio and anisotropy coefficient to achieve an identical mathematical solution of drainage. An example where identical mathematical solutions for drainage are obtained is provided in Figure 11, where the bottom plot is obtained by taking the properties of the top plot and dividing the aspect ratio by 8, dividing the anisotropy coefficient by $8^2$, and dividing $\kappa$ by 8. While the solutions are mathematically identical, note that

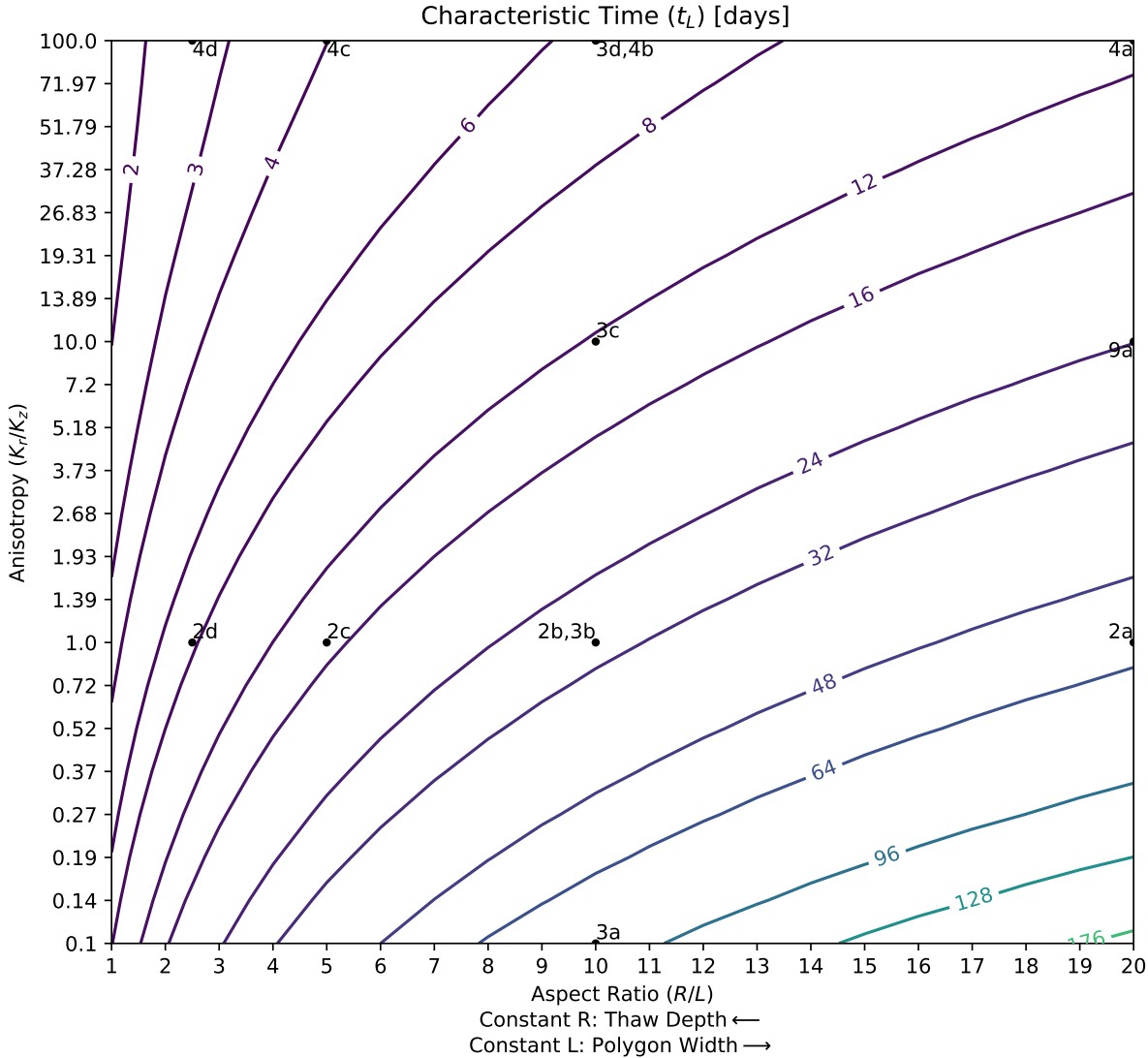

**Figure 10.** Contour map of the characteristic time of polygon drainage as a function of polygon aspect ratio and anisotropy coefficient. Locations associated with other figures are indicated by labeled points. By choosing a constant value for $R$ $(L = R/x)$, horizontal transects parallel to the x-axis represent the effect of thaw depth increasing from right to left, while they represent increasing polygon width from left to right for constant values of $L$ $(R = Lx)$. Note that $\kappa = 5$ day$^{-1}$ in these calculations, and therefore, the location associated with Figure 11b cannot be indicated.

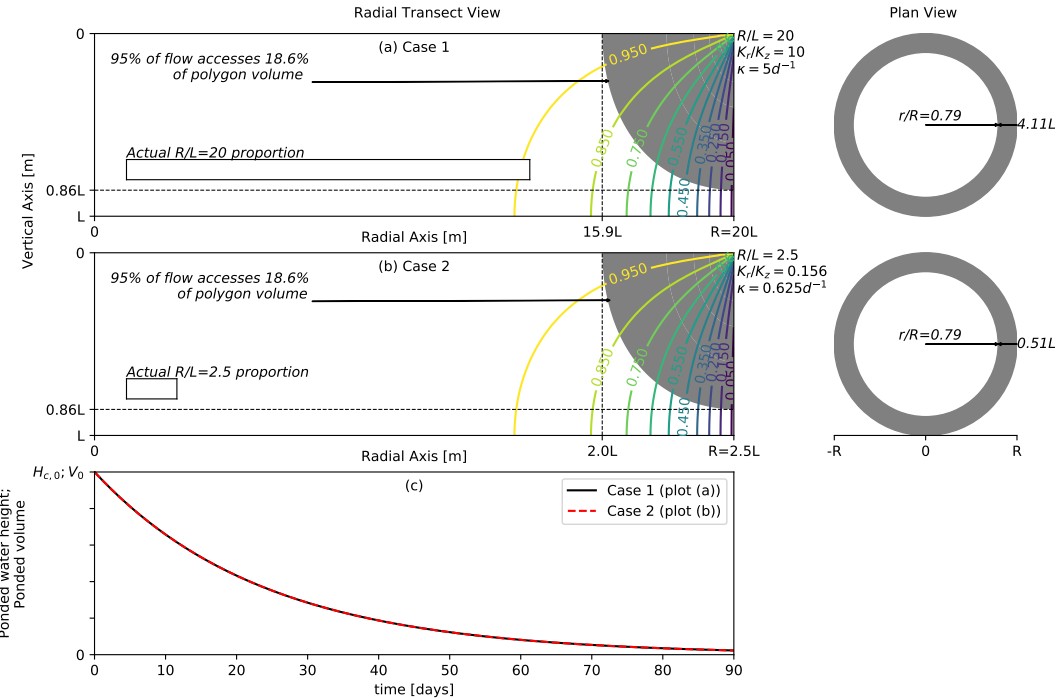

**Figure 11.** Attaining the same mathematical solution by modifying polygon aspect ratio ($R/L$), anisotropy coefficient ($K_r/K_z$), and rim conductance ($\kappa$). The left plots of (a) and (b) contain polygon radial transect head contours (colored lines) and filled stream tubes (grey regions) for several polygon aspect ratios (radius/thickness). The grey shaded region denotes the portion of the transect accessed by 95% of the flow. The right plots of (a) and (b) contain corresponding grey rings indicating the surface area where 95% of the polygon flow infiltrates. Plots (a) and (b) contain a rectangle drawn to the actual proportions for the given polygon aspect ratio. The ponded water height/volume depletion curves for both cases are presented in (c).

the axes in both plots are not to scale (actual proportions are provided as insets to the plots) and that there are differences in some relative dimensions indicated in the figure (for example, the relative width of the polygon volume accessed by 95% of the drainage flow is around $4L$ in the top case and around $0.5L$ in the bottom case).

In the bottom plot in Figure 11, we demonstrate that despite these differences, the depletion curves are identical for both cases. This result is due to our choice to modify $R$ to change the aspect ratio and modify $K_r$ to change the anisotropy coefficient. If either $L$ or $K_z$ are chosen to modify the aspect ratio or anisotropy coefficient, respectively, the depletion curves would not necessarily be identical. For example, if $L$ is multiplied by 8, $K_z$ multiplied by $8^2$, and $\kappa$ divided by 8 (equivalent modifications to aspect ratio and anisotropy coefficient as above), the drainage flow net would still be mathematical equivalent, but the characteristic time would be 8 times shorter than in the original case (refer to equation A11). Therefore, while it is possible

to obtain mathematically equivalent drainage patterns with counteracting modifications to aspect ratio, anisotropy coefficient, and outflow conductance, the temporal depletion will only be equivalent if $R$ and $K_r$ are used to modify the aspect ratio and anisotropy coefficient, respectively.

## 4    Discussion

Our analysis provides new insights into the relative effects of geometry and anisotropy on the manner in which inundated ice-wedge polygons retain and slowly release water from their centers to their troughs, which form the drainage network of polygonal tundra landscapes. Using a mathematical representation of inundated ice-wedge polygon drainage (Zlotnik et al., 2020) based on extensive field observations (Helbig et al., 2013; Koch et al., 2014; Liljedahl and Wilson, 2016; Wales et al., 2020) and verified here through calibration to field measurements, we quantify the sensitivity of inundated ice-wedge polygon drainage to representative polygon sizes, inter- and intra-annual changes in thaw depth, and preferential flow (hydraulic conductivity anisotropy).

### 4.1    Calibration implications

The calibration identifies factors which need to be considered by any hydrologic model to simulate drainage from an inundated polygon center. Using a parsimonious model, we were able to identify the refinements required in a polygon drainage model to capture center water levels. Using more complex models would likely obfuscate the identification of these refinements. The final model formulation provides a fast model for predicting the manner and timing of polygon drainage driven by environmental factors. The calibration of the model, driven by transient boundary conditions, provides confidence in the real-world applicability of the sensitivity analysis based on comparisons of steady-state snapshots of the model.

The first refinement is a precipitation multiplier and is based on a simple mass balance indicating that the measured precipitation cannot account for the total increase in ponded water levels after precipitation events. The precipitation multiplier accounts for the fact that precipitation will run off from the rims into the center pond resulting in a larger increase in ponded water than precipitation measured by a rain gauge. The precipitation multiplier also accounts for any potential rain gauge under catch. The second refinement is a thaw-depth dependent vertical hydraulic conductivity that accounts for the decreased hydraulic conductivity of deeper soil layers. This refinement accounts for the decrease in vertical hydraulic conductivities as the thaw depth increases and includes less hydraulically conductive soils. The final calibration provides a good match to measurements, with a sub-centimeter RMSE of 0.37 cm and an $R^2$ (equivalent to the Nash-Sutcliffe Efficiency here) of 0.96. A key insight from our research is that polygon drainage models need to consider decreases in effective vertical hydraulic conductivity with increasing thaw depth. This research enhances our hydrologic understanding of an Arctic landscape that is undergoing rapid transition due to a warming climate. The hydrology of these landscapes has important implications concerning carbon transport and emissions, subsidence, and Arctic shrubification, to name a few Arctic processes of concern.

Due to covariance in the effects of the hydraulic conductivity parameters on water levels, the hydraulic conductivity parameters are loosely constrained by the calibration (based on local sensitivities). This indicates the importance of constraining

the hydraulic conductivity parameters with field measurements if possible (field measurements are not available in our case). However, despite relatively large standard errors in the hydraulic conductivity parameter estimates, the calibration identifies physically realistic values. It should also be noted that despite vertical hydraulic conductivity being loosely constrained in the final calibration (calibration case 3), the model was unable to match water levels with a constant vertical hydraulic conductivity (calibration case 2). This indicates the importance of the thaw-depth dependent vertical hydraulic conductivity in the model. The other parameters (discharge conductance, initial polygon-center water level, and precipitation multiplier) are all well constrained by the analysis.

The water level (Liljedahl and Wilson, 2016) and temperature (Romanovsky et al., 2017) measurements are well constrained with high degrees of resolution and accuracy. While the resolution of precipitation measurements is high (0.1 mm), there is the potential for under catch during windy precipitation events. More importantly with regard to ponded water levels, the precipitation measurements do not account for the runoff of water during precipitation events from rims into the polygon center, resulting in measured precipitation less than increases in pond water levels. This uncertainty is accounted for through the calibration of a precipitation multiplier, which effectively captures the effect of runoff from the polygon *watershed* into the center pond along with any potential under catch during windy precipitation events.

## 4.2 Analysis implications

A key result of this study is that the geometry and anisotropy of the polygon subsurface have a significant effect on the region of the polygon subsurface predominantly accessed by drainage flow and the time to transition from inundated to drained. Due to the geometry of inundated polygons, the primary drainage pathway is restricted to an annular, radially-peripheral region of the ice-wedge polygon center near the rim. As a result, the middle and lower portions of the polygon center are excluded from the majority of the drainage to varying degrees depending on the polygon geometry and anisotropy. Also, the majority of the ponded water will flow towards the rim of the polygon before infiltrating into the subsurface. Field observations have indicated not only the existence of intra-polygon biogeochemical diversity (Zona et al., 2011; Newman et al., 2015), but that, based on mineral and nutrient loading, pervasive subsurface flow from centers to troughs exists (Koch et al., 2014). Based on our analysis, polygon geometry and anisotropy will have important effects on the biogeochemistry of polygons (Heikoop et al., 2015; Throckmorton et al., 2015; Newman et al., 2015; Wales et al., 2020; Plaza et al., 2019) affecting dissolved organic matter and mineral flushing from the subsurface of polygon centers to the surface waters of troughs. This is important not only simply concerning the discharge of dissolved organic matter from polygonal tundra landscapes, but also for the biogeochemical effects due to photolysis of dissolved organic matter newly exposed to sunlight (Laurion and Mladenov, 2013; Cory et al., 2014).

The potential that the annular, radially-peripheral region near the rims will be well flushed of nutrients, while the middle may not, indicates the need for additional field studies designed to measure the effects of anisotropy and preferential flow paths on thermal-hydrology and biogeochemistry. For isotropic cases, it should also be considered that the drainage will spread out further towards the middle of the polygon center as the thaw season progresses and the thawed soil layer thickens (in other words, the aspect ratio decreases; Figures 4 and 6). However, with high anisotropy coefficient, the drainage still spreads out towards the polygon middle as the thaw season progresses (aspect ratio decreases), but the depth of the main drainage pathways

contract towards the ground surface (Figure 6). Ultimately, the effect of this vertical contraction outweighs the lateral spreading, leading to a smaller region being accessed by drainage. Therefore, in the case of smaller and/or more deeply thawed (low aspect ratio) polygons with high anisotropy coefficient, the region close to the surface will be flushed more than the deeper regions.

Deeper nutrients that become available for transport because of recent thaw may still be less accessible to aqueous transport due to drainage pathways in polygons with high anisotropy coefficient. Field measurements of geochemical depth profiles in polygons support limited deep flushing. As reviewed in Newman et al. (2015), concentration increases of geogenic solutes and organic carbon with active layer depth are relatively common in polygonal terrain. It is unlikely that these sometimes large geochemical depth gradients would be preserved with significant amounts of deep flushing. Inter-annual deepening of

the active layer will lead to even more dramatic evolution in drainage patterns described above during the thaw season as the range of thaw depths encountered increases. This research reinforces the need for field studies on anisotropy and preferential flow in polygon landscapes to better understand the hydrologic transitions and feedbacks that will occur in a warming climate.

    For a given thaw depth, advective heat transport will be more focused near the rim for larger polygons, and may result in enhanced ice-wedge degradation (Wright et al., 2009). Based on the relative drainage pathways, advective heat transport will

be most pronounced in larger and/or shallower (large aspect ratio) polygons with low anisotropy coefficient (refer to Figures 4a and 5a). Therefore, drainage events for wide, isotropic (or with preferential vertical flow) polygons may result in enhanced ice-wedge top thawing, which would promote low- to high-centered polygon transition. The more spread out the drainage is throughout the thawed soil layer, the less pronounced this effect may be (for example, wide polygons with high anisotropy coefficient as in Figures 6a and b). However, small polygons with high anisotropy coefficient will restrict the drainage flow

near the ground surface, potentially reducing the advective transport of heat to the ice-wedge top as well (Figure 6d). Similarly, increasing seasonal thaw depth and inter-annual active layer thickness will lead to polygons with less focused advective heat transport towards ice-wedge tops, potentially providing a negative feedback on ice-wedge degradation. These results provide an additional perspective on ice-wedge degradation, complementing previous research that found that the process is also strongly controlled by geometry (Abolt et al., 2020) and hydrologic conditions (Nitzbon et al., 2019).

Small polygons with deeply thawed soil layers (low aspect ratios) and high horizontal preferential flow (high anisotropy coefficient) have the potential to drain most quickly. Therefore, all other factors being equal, regions of polygonal tundra characterized by small, deeply thawed, anisotropic polygons will drain more quickly and consequently will have a greater potential for nutrient flushing, transition from methane to carbon dioxide atmospheric emissions, and biological succession than regions with large, shallowly thawed, isotropic polygons. Concerning temporal changes during the thaw season, polygon

pond depletion will slow down as the thaw depth increases, and this reduction will become more dramatic over the thaw season as inter-annual active-layer thickness increases. This implies that a thickening active layer due to warming trends may result in slower pond depletion. For a given location, factors such as regional flow patterns, large-scale topography, etc., will influence the region's overall drainage timing. However, along with these other factors, our analysis indicates that aspect ratio will have a nearly linear positive relationship while the anisotropy coefficient will have an exponential negative relationship with drainage

timing.

## 4.3 Model limitations

The drainage pathways and timing presented here are based on hydrogeological first principles (Harr, 1962; Cedergren, 1968; Freeze and Cherry, 1979; Bear, 1979) using a generalization of ice-wedge polygon geometry and hydraulic properties that allow insights into drainage pathways and timing. The model captures the physical forces that ponded water exerts on a cylindrical porous disc with drainage allowed radially through its sides. While the cylindrical geometry and anisotropy coefficients considered here do not cover all potential variations present in ice-wedge polygons, the impact of those variations will cause deviations from the idealized scenarios considered here. For example, heterogeneities will warp the base case hydraulic head equipotentials and streamlines. The cylindrical idealization of ice-wedge polygon geometry will best approximate drainage in nearly symmetrical hexagonal ice-wedge polygons, while non-symmetric or square ice-wedge polygon drainage will deviate most from that shown here. The model assumes uniform radial discharge around the periphery of the polygon center. Low points in the elevated frozen ground under the rim will lead to deviations from our results. Elevated frozen ground under the rim will also warp the streamlines upwards within a localized region near the rim, but this effect will dissipate a short distance into the polygon center.

In our sensitivity analysis, we use idealized conceptualizations and dimensionless variables which allow hydrologic characteristics of drainage to be exposed in their most fundamental form. To clearly and concisely expose these characteristics, we neglect factors such as evaporation, precipitation, and non-idealized polygon geometry (evaporation and precipitation are included in the calibration). As a result, our analysis is not intended to provide predictive capability across all polygonal tundra scenarios, but to provide hydrologic intuition into the relative effects of geometry and anisotropy on inundated ice-wedge polygon drainage and timing.

As the analytical solution applies to ponded conditions, it does not apply to freeze-up at the end of the thaw season after the ponded water in the polygon center freezes. At this time, the active layer freezes simultaneously from the top and bottom and cryosuction draws water towards the freezing fronts. This redistribution of water affects ice-wedge polygon drainage and Wales et al. (2020) postulate that it may explain some of the observations of their tracer test. More complex models than applied here are required to capture the details of water redistribution during freeze-up (Painter, 2011; Atchley et al., 2016; Schuh et al., 2017).

Since the model is based on the saturated groundwater flow equation, in its current form it cannot be applied to non-inundated low-centered polygons. Since it is based on having a ponded center, the model is also not applicable to high-centered polygons. However, a similar approach to that presented here could be taken with the unsaturated groundwater flow equation to capture these other polygon scenarios and types.

## 4.4 Relation to other mathematical analyses

As our analysis is the first detailed examination of inundated ice-wedge polygon drainage patterns, our results provide a new perspective to existing mathematical analyses. Much of the existing mathematical analyses investigate drainage from polygonal landscapes. For example, Cresto-Aleina et al. (2013) analyze low-centered polygonal landscape hydrology stochastically using

percolation theory, a method that is not designed to investigate drainage pathways of individual polygons. Nitzbon et al. (2019, 2020) investigate the hydrology of polygonal tundra using a "tiled" approach that is also not designed to map individual ice-wedge polygon drainage pathways. In these cases, our analysis enhances our understanding by considering details that are glossed over in these analyses for the sake of considering drainage from many polygons. Other research has focused on process-rich 1D analyses that are unable to consider lateral flow. For example, Atchley et al. (2015); Harp et al. (2016) and Atchley et al. (2016) use a process-rich, complex 1D model to calibrate and perform uncertainty and sensitivity analyses of thermal-hydrology models of ice-wedge polygons. Our analysis augments these by providing insights into lateral fluid flow. Abolt et al. (2020) perform a sensitivity analysis of the thermal effects of ice-wedge thaw on individual polygons using a full-physics thermal-hydrology model. Our analysis adds to this research by providing indications of the drainage patterns that would exist along with the thermal effects. Jan et al. (2020) calibrate a full-physics thermal-hydrology model of ice-wedge polygons using field data, but since the focus of their research is calibration, they do not investigate drainage pathways. As a result, our analysis provides new information that augments existing research, helps the Arctic hydrology research community gain an intuitive understanding of inundated ice-wedge polygon drainage, and provides new modeling and experimental research directions based on the idealized base cases considered here.

## 5 Conclusions

Our results affirm that existing conceptualizations of polygon drainage need revisiting, as was implied by a preliminary analysis conducted by Zlotnik et al. (2020). Fundamental hydrological principles indicate that, due to low-centered polygon geometry and hydraulic properties, inundated polygon drainage flux will not be uniform throughout the thawed soil layer. This result indicates that while some portions of the polygon thaw layer will be well flushed, other portions will be poorly flushed. This has important implications for the aqueous transport of carbon and other dissolved nutrients from polygonal tundra. It also indicates the potential for focused advective heat transport from the ponded water towards the ice-wedge tops, which could enhance ice-wedge degradation. Within this context, we provide the following conclusions:

- A simple analytical solution based on hydrological first principles is able to capture ice-wedge polygon drainage dynamics over an entire thaw season. Consideration within the model of runoff from topographic highs (rims) during rain events and decreasing effective vertical hydraulic conductivity with increasing thaw depth are required to match field measurements. We were able to identify these necessary model components by using a parsimonious model that has the ability to fail.

- We provide rigorous confirmation that the majority of drainage from inundated ice-wedge polygon centers occurs along an annular region along their radial periphery; however, polygon geometry and hydraulic conductivity anisotropy significantly impact the drainage pathways, as originally postulated by Zlotnik et al. (2020). This implies that nutrient flushing and the advective thermal transfer will be focused along polygon edges, neglecting the polygon center. Additionally, our results indicate that:

- A combination of high aspect ratio (wide, shallow polygons) and high anisotropy coefficient (preferential horizontal flow) results in the greatest spreading of drainage flow towards the middle of the polygon center and the largest fraction of the polygon volume being accessed by drainage flow. In these cases, the nutrient flushing will be more uniform than other cases and advective heat transport towards the ice-wedge top will be less focused and therefore less able to thaw the ice-wedge top.

- A combination of high aspect ratio (wide, shallow polygons) and low anisotropy coefficient (preferential vertical flow) results in the greatest focusing of drainage flow and the smallest fraction of the polygon volume being accessed by drainage flow. In these cases, nutrient flushing will be more localized along the outer edge of the polygon (non-uniform) and the advective heat transport towards the ice-wedge tops most focused, possibly resulting in greater degradation of ice-wedges than in other cases.

- Combinations of aspect ratio and anisotropy coefficient have counteracting effects of radial versus vertical extension/contraction of drainage pathways, producing non-monotonic relationships between aspect ratio/anisotropy coefficient and accessed volume (curved feature in accessed volume response surface in Figure 7). Therefore, the combined, non-linear effects of geometry and anisotropy on drainage patterns must be considered when evaluating nutrient flushing and advective heat transport.

- Polygon drainage time has an approximately positive linear relationship with aspect ratio when anisotropy is held constant; in other words, wide, shallow polygons drain slowly while small, deep polygons drain quickly. This implies that polygonal tundra with larger polygons may drain more slowly than tundra composed of smaller polygons.

- Polygon drainage time has a negative exponential relationship with anisotropy coefficient when aspect ratio is held constant. In other words, increases in preferential horizontal flow lead to exponentially faster drainage. Therefore, polygonal tundra with greater preferential horizontal flow, due to more pronounced horizontal stratigraphy or ice lens development, will drain faster, while less horizontally stratified tundra, due to cryoturbation, for example, will drain more slowly.

*Code availability.* Matlab code of the analytical solution is available via Zlotnik et al. (2020) at http://www.mdpi.com/2073-4441/12/12/3376/s1.

*Video supplement.* A video illustrating the validation of the analytical solution to an inundated ice-wedge polygon drainage event in 2012 near Utqiaġvik is available via Zlotnik et al. (2020) at http://www.mdpi.com/2073-4441/12/12/3376/s1.

## Appendix A

Here, we present the analytical solutions for hydraulic heads and stream function under the center of an inundated ice-wedge polygon and the depletion curve of the ponded water height due to drainage to the surrounding trough. For details on the derivations and validation of the model, refer to Zlotnik et al. (2020).

### A1  Hydraulic head and stream function analytical solutions

We idealize the subsurface below the center region of a low-centered polygon as a thin cylinder with radius in the horizontal direction and length in the vertical direction (refer to Figure 1). Based on this approximation, the hydraulic heads ($h(r, z, t)$) in an inundated polygon will satisfy the following equation:

$$\frac{K_r}{r}\frac{\partial}{\partial r}\left(r\frac{\partial h}{\partial r}\right) + K_z\frac{\partial^2 h}{\partial z^2} = S_s\frac{\partial h}{\partial t}. \tag{A1}$$

However, considering that storativity is negligible given the vertical and horizontal scale of the domain, the right-hand term can be neglected as

$$\frac{K_r}{r}\frac{\partial}{\partial r}\left(r\frac{\partial h}{\partial r}\right) + K_z\frac{\partial^2 h}{\partial z^2} = 0, \tag{A2}$$

where $h$ is the hydraulic head, $r$ is the radial coordinate, $z$ is the depth coordinate (positive in the downward vertical direction), $K_r$ is the horizontal (radial) hydraulic conductivity, $K_z$ is the vertical hydraulic conductivity, $t$ is time, and $S_s$ is specific storage. The system is fully specified by ensuring that (BC1) the heads along the central vertical axis ($h(0, z, t)$) are finite, (BC2) the heads along the ground surface are equal to the height of ponded water in the polygon center ($h(r, 0, t) = H_c(t)$), (BC3) the change in heads along the outer vertical boundary is governed by the change in heads along the boundary and the height of water in the trough ($H_t$)

$$-K_r\frac{\partial h(R, z, t)}{\partial r} = \kappa(h(R, z, t) - H_t), \tag{A3}$$

(BC4) the bottom of the model has a zero head gradient

$$\frac{\partial h(r, L, t)}{\partial z} = 0,\ 0 < r < R, \tag{A4}$$

and (BC5) the change in volume of ponded water in the polygon center is related to the vertical head gradients along the ground surface,

$$\pi R^2\frac{dH_c(t)}{dt} = 2\pi K_z\int_0^R\frac{\partial h(r, 0, t)}{\partial z}r dr,\ H_c(0) = H_{c,0}, \tag{A5}$$

where $R$ is the radius of the polygon, $L$ is the depth of the thawed subsurface of the polygon, $H_c(t)$ is the ponded water height in the center of the polygon at time $t$, $H_t$ is the height of water in the trough, and $\kappa$ characterizes the conductance to flow across the drainage interface to the trough (in our case, the hydraulic conductance of the soil layer under the rim).

Using dimensionless coordinates and parameters defined as

$$r^* = \frac{r}{L}\sqrt{\frac{K_z}{K_r}}, \quad z^* = \frac{z}{L}, \quad R^* = \frac{R}{L}\sqrt{\frac{K_z}{K_r}}, \quad \text{Bi} = \frac{\kappa L}{\sqrt{K_r K_z}}, \tag{A6}$$

dimensionless solutions for hydraulic heads and the stream function can be obtained as

$$h^*(r^*, z^*) = \frac{h(r,z,t) - H_t}{H_c(t) - H_t} = 2\sum_{n=1}^{\infty} \frac{J_1(\lambda_n R^*)}{\lambda_n R^* \left[ J_0^2(\lambda_n R^*) + J_1^2(\lambda_n R^*) \right]} \frac{\cosh(\lambda_n(1-z^*))}{\cosh(\lambda_n)} J_0(\lambda_n r^*) \tag{A7}$$

and

$$\Psi^*(r^*, z^*) = \frac{\psi(r^*, z^*)}{\psi(R^*, 0)} = 2\sum_{n=1}^{\infty} \frac{J_1(\lambda_n R^*)J_1(\lambda_n r^*)r^*}{\lambda_n R^* \left[ J_0^2(\lambda_n R^*) + J_1^2(\lambda_n R^*) \right]} \frac{\sinh(\lambda_n(1-z^*))}{\cosh(\lambda_n)}, \tag{A8}$$

respectively, where $J_m$, with $m$=0 or 1, is the Bessel function of the first kind of $m$th order and $\lambda_n$ is the $n$th root of the

630 equation

$$\lambda_n J_1(\lambda_n R^*) = \text{Bi} J_0(\lambda_n R^*), \ n = 1, 2, \dots \tag{A9}$$

The solutions can be verified by direct substitution of equations A7 and A8 into the boundary value problem defined by equations A2 to A6.

## A2 Ponded height depletion curve analytical solution

The ponded height depletion curve can be defined as

$$H_c(t) = H_t + [H_{c,0} - H_t]e^{-t/t_L}, \tag{A10}$$

where $t_L$ is the characteristic depletion time defined as

$$t_L = \frac{R^2}{2K_r L F(\text{Bi}, R^*)}, \tag{A11}$$

which is the time when $H_c(t)/H_{c,0} = 1/e$; in other words, when the ponded height is approximately 37% of its initial. The

640 function $F(\text{Bi}, R^*)$ can be evaluated as

$$F(\text{Bi}, R^*) = \int\limits_0^{R^*} \frac{\partial h^*(r^*, 0)}{\partial z^*} r^* dr^* = 2\sum_{n=1}^{\infty} \frac{\tanh(\lambda_n)}{\lambda_n \left[ (\lambda_n/\text{Bi})^2 + 1 \right]}. \tag{A12}$$

The depletion curve can be expressed in non-dimensional terms as

$$H^*(t^*) = \frac{H_c(t) - H_t}{H_{c,0} - H_t} e^{-t^*}, \tag{A13}$$

where non-dimensional time $t^* = t/t_L$. The solution can be verified by direct substitution of equation A10 into equation A5.

*Author contributions.* DH and VZ developed the conceptual model of inundated polygonal tundra hydrology. VZ derived the analytical solutions. VZ and DH encoded the analytical solutions. CA created Figure 1. DH created Figures 4-11. AA provided text for the Introduction and Discussion sections. EJ provided text for the discussion section. AA, BN, EJ, and CW provided critical reviews of the manuscript. CW secured funding.

*Competing interests.* The authors declare no competing interests.

*Acknowledgements.* The Next Generation Ecosystem Experiments Arctic (NGEE-Arctic) project (DOE ERKP757), funded by the Office of Biological and Environmental Research within the U.S. Department of Energy's Office of Science supported this research. Sofia Avendaño and Bulbul Ahmmed provided reviews during the development of this article providing technical and editorial improvements.

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
