# Peer review of "New insights into the drainage of inundated ice-wedge polygons using fundamental hydrologic principles"

_The Cryosphere, 2020_

## Author Comment (AC1)

Response to RC1 for tc-2020-351

RC1 – black text; Response – blue text

In the present paper the authors investigate a novel analytical model which conceptualizes the hydrological drainage dynamics of inundated ice-wedge polygon centers in Arctic lowlands. I have already reviewed an earlier submission of this manuscript with TC (https://tc.copernicus.org/preprints/tc-2020-100/tc-2020-100-RC1.pdf). One of my major concerns with the previous submission was, that the article by Zlotnik et al. (2020) which introduces the analytical model had not undergone peer-review at the time of the initial submission. This point is now obsolete, as the model description and validation article has been published in a peer-reviewed journal (http://www.mdpi.com/2073-4441/12/12/3376).

The present article is well-written, well-presented, and certainly of interest to the readers of TC as the topic is important and timely. In particular, I have seen that the authors addressed almost all major and minor points which I raised when reviewing the first submission of this article. Hence, I support publication of this article in TC after addressing some minor, mostly technical, points which I noticed during reading.

We thank Dr. Nitzbon for taking the time to review our manuscript for a second time. We are glad that Dr. Nitzbon feels that we adequately addressed most major and minor points from his first review. We found his first review extremely helpful and constructive in improving our manuscript. Please find our responses below indicating how we have further revised the manuscript based on Dr. Nitzbon's additional comments.

**Specific comments**

- The introduction of the article is quite long and should be shortened and streamlined. For instance, I would suggest to shorten on detailed descriptions and justifications which should rather be presented in the Methods or Discussions sections, respectively (e.g. lines 84-87, lines 107-110, lines 112f).

We agree and have removed the indicated lines. The Robin boundary condition discussion (lines 84-87) has been integrated into "Section 2.1 Model overview" and the precipitation/evaporation discussion (lines 107-110) has been moved to "Section 2.2 Model parameterizations". The material in lines 112f is covered adequately elsewhere (such as in "Section 4.2 Model limitations"), so has simply been removed.

- In the abstract (l. 7) the authors state to investigate ``inter-annual increases'' in active layer thickness. While I understand that this is only done indirectly via the variation of aspect ratios, it would be nice to provide a discussion of the effect of active-layer deepening, similar to what is done for the seasonal thaw-depth increase in lines 323ff.

We agree that tying back into the "inter-annual increases in active layer thickness" was missing from the Discussion Section. As Dr. Nitzbon indicates, we use aspect ratio to evaluate both the effects of seasonal thaw depth and inter-annual active-layer thickness changes. Therefore, the

discussion in 323ff applies to both as well. To make this more apparent, we have added the following text:

"Inter-annual deepening of the active layer will lead to even more dramatic evolution in drainage patterns described above during the thaw season as the range of thaw depths encountered increases."

We also point this out to the reader concerning advective heat transport towards ice-wedge tops:

"Similarly, increasing seasonal thaw depth and inter-annual active layer thickness will lead to polygons with less focused advective heat transport towards ice-wedge tops, potentially providing a negative feedback on ice-wedge degradation."

Concerning pond depletion rate, we have added the following text:

"Concerning temporal changes during the thaw season, polygon pond depletion will slow down as the thaw depth increases, and this reduction will become more dramatic over the thaw season as inter-annual active-layer thickness increases. This implies that a thickening active layer due to warming trends may result in slower pond depletion."

These additions to the discussion help guide the reader to the implications of our results, and we thank Dr. Nitzbon for prompting them.

**Technical corrections**

- l. 2: I think ``transitions from methane to carbon dioxide dominated emissions'' would describe the implication of polygon drainage better.

We completely agree and have revised the text.

- l. 68:``ice-wedge surface hydrology'' might be a confusing terminology. Maybe rephrase this to ``(ice-wedge) polygon surface hydrology'' or ``polygonal tundra surface hydrology''.

Dr. Nitzbon is exactly correct that the wording was confusing and imprecise. We have used his suggestion and changed this to "polygonal tundra surface hydrology".

- l. 319: Do you mean ``affecting''?

Yes! We thank Dr. Nitzbon for pointing this out. It has been fixed.

- l. 345: Do you mean Abolt et al. (2020) (JGR: Earth Surface), which I suggested to discuss in the first review? To my understanding Abolt et al. (2018) do not discuss the effect of trough geometry.

We agree that Abolt et al. (2020) is a very appropriate reference here. We thank Dr. Nitzbon for pointing out that oversight and apologize for not catching that suggestion in his first review.

- l. 385: should be ``that'' instead of ``this''

This has been fixed.

- The references Atchley et al. (2015) and Harp et al. (2015) are for the Discussion papers, but not for the final revised articles. You probably want to change this.

We thank Dr. Nitzbon for catching this. The references have been updated to the final articles.

- Fig. 5 and 8: Consider leaving away the decimal points (.000) at the contour line labels.

Yes, the ".000"s were completely unnecessary and we have removed them. We appreciate Dr. Nitzbon pointing this out.

---

## Author Comment (AC2)

We thank the reviewer for a thoughtful and critical review of our manuscript. The reviewer's comments have allowed us to:

- Provide clear delineation between our conclusions and those in Zlotnik et al. (2020), explicitly stating where we have provided rigorous confirmation, elaboration, and extension, of existing conclusions and where we have provided completely new conclusions.
- Clearly describe the potential impacts of a raised thaw table under the rims based on fundamental hydrology using the well-known hydrologic analogy of a partially penetrating well in an aquifer.
- Demonstrate the ability of the model to account for heterogeneous soil layering using an effective vertical hydraulic conductivity in a calibration extending over an entire thaw season.
- Make the paper more interpretable by changing model specific language to plain language, as well as generally improve the grammar and readability.
- Provide physical interpretations of our range of anisotropies.

The reviewer's comments have led to a major revision of the manuscript and entail a significant improvement, for which we are grateful. Please refer to our responses below for more details on these improvements. The reviewer's comments are included below in black while our responses are in blue.

The manuscript "New insights into the drainage of inundated ice-wedge polygons using fundamental hydrologic principles" presents evaluations of tundra polygon drainage characteristics with a simple analytical model. While I like the idea of simplified modelling to evaluate the properties of the polygon hydrological system, the key findings seem to be identical or at least close to the already published study by Zlotnik et al. (2020), even if the quantitative analysis is different. If the authors maintain that the manuscript contains novel research, they need to explain the relationship between the two studies much better. If the qualitative conclusions are indeed largely the same and the main novelty of this work is additional quantitative scenarios and model evaluation, the authors need to consider and discuss the model limitations in much more detail. From my limited understanding of the model, I have the impression that it cannot describe many relevant real-world situations, at least not quantitatively. In conclusion, the authors need to carefully argue what in their study is novel at a level that would warrant publication in TC.

Our manuscript rigorously confirms, fully fleshes out, and expands on Zlotnik et al. (2020). Additionally, we have now added a calibration to season-long field measurements to the manuscript building confidence in the utility of the model and identifying necessary model refinements in order to match observed water levels.

We agree that there are similarities between the conclusions of our manuscript and Zlotnik et al. (2020). The papers were originally written as companion papers, where Zlotnik et al. introduces the model and provides some preliminary scoping calculations. The current manuscript builds on those preliminary indications of polygon drainage dynamics, rigorously confirming many of its conclusions, and identifying many nuances in the model sensitivities not identified in Zlotnik et

al. (2020) (e.g., nuances identified in the full analysis of the parameter space in Figures. 6 and 9). As the papers did not end up as companion papers, we agree that the link between the two needs to be more explicit. We now do this in the abstract:

"In this research, we perform (1) a rigorous model sensitivity analysis that expands on previously published indications of polygon drainage characteristics and (2) a calibration to field data identifying necessary model refinements."

The introduction:

"In this paper, we use a recently developed model (Zlotnik et al., 2020) based on fundamental hydrogeological principles…"

And the conclusions:

"Our results affirm that existing conceptualizations of polygon drainage need revisiting, as was implied by a preliminary analysis conducted by Zlotnik et al. (2020)."

and

"We provide rigorous confirmation that the majority of drainage from inundated ice-wedge polygon centers occurs along an annular region along their radial periphery; however, polygon geometry and hydraulic conductivity anisotropy significantly impact the drainage pathways, as originally postulated by Zlotnik et al. (2020)."

And now clearly differentiate, throughout the manuscript, between the preliminary scoping analysis in Zlotnik et al. (2020) and our rigorous sensitivity analysis and new conclusions.

Additionally, we have added the results of a season-long calibration that verifies the ability of the model to capture drainage dynamics and identifies the minimum model refinements necessary to do this. This not only provides confidence in the utility of the model, but also provides additional insights into the hydrology of polygon drainage not presented before.

Major Comments:

1.  The central conclusions of the manuscript, as stated in the Abstract, appear to be largely identical with the ones in Zlotnik et al. (2020).

This manuscript (Abstract): "One of the primary insights from the model is that most inundated ice-wedge polygon drainage occurs along an annular region of the polygon center near the rims. This implies that inundated polygons are most intensely flushed by drainage in an annular region along their horizontal periphery, with implications for transport of nutrients (such as dissolved organic carbon) and advection of heat towards ice-wedge tops."

In plain language: Drainage and flushing of the center is concentrated to the area adjacent to the rim. This (qualitative statement only) affects water-mediated transport.

Zlotnik et al., 2020 (Conclusions): "only a small fraction of the polygon volume near the rim area is flushed by the drainage at relatively high velocities, suggesting that nearly all advective transport of solutes, heat, and soil particles is confined to this zone."

In plain language: Drainage and flushing of the center is concentrated to the area adjacent to the rim. This (qualitative statement only) affects water-mediated transport.

This manuscript (Abstract): "The model indicates that polygons with large aspect ratios and high anisotropy will have the most distributed drainage. Polygons with large aspect ratios and low anisotropy will have their drainage most focused near their periphery and will drain most slowly. Polygons with small aspect ratios and high anisotropy will drain most quickly."

In plain language: For a given fixed polygon radius, and for a given fixed vertical hydraulic conductivity: increasing the horizontal conductivity increases drainage, and increasing the thaw depth increases drainage as well. Both also lead to a less focused flow within the center, i.e. flushing by throughflow of water occurs over a larger volume of the center.

Zlotnik et al., 2020 (Conclusions): "Anisotropy in hydraulic conductivity (horizontal-to-vertical hydraulic conductivity ratio) has a secondary influence on the intensity of flushing. Increases of anisotropy values counteract the effects of increased geometrical aspect ratio increases and vice versa."

Zlotnik et al., 2020 (Appendix B): "…an increase in the anisotropy can redistribute the flux over the polygon, thereby reducing the edge effect."

In plain language: For a given fixed polygon radius, and for a given fixed vertical hydraulic conductivity, increasing the horizontal conductivity has the same qualitative effect on drainage as increasing the thaw depth. An increase in horizontal conductivity leads to a less focused flow within the center, i.e. flushing by throughflow of water occurs over a larger volume of the center.

The first conclusion seems to be identical, and the second one is very close, although stated more clearly in this work. Worryingly, the authors do not make an attempt to acknowledge this similarity and to explain the differences between the two studies to the reader. Zlotnik et al. (2020) is only presented briefly as a model description paper, without discussing the relation between the two studies. I can see that the present manuscript contains additional and more quantitative analysis of the model trajectories. However, I have the impression that it is largely an illustration and a more detailed description of the main findings published in Zlotnik et al. (2020).

We appreciate that the reviewer recognizes that the current manuscript "contains additional and more quantitative analysis of the model trajectories". We agree that the first conclusion is identical. However, Zlotnik et al. (2020) provided preliminary indications of this phenomenon, which is rigorously confirmed, thoroughly evaluated, and expanded upon in the current manuscript. The second conclusion also builds on the preliminary analysis in Zlotnik et al. (2020), however, significantly expands upon it through a thorough sensitivity analysis providing

insights not included in Zlotnik et al. (2020) (e.g., Figures 2-5). Our manuscript presents new information at a level of detail far beyond the preliminary results presented in Zlotnik et al. (2020). In particular, Figures 5 and 8 significantly expand on the insights drawn from the model compared to what is presented in Zlotnik et al. (2020). We agree that this point needs to be stated more clearly, and we now clearly differentiate between the findings in Zlotnik et al (2020) and the current manuscript. Please see details and examples in our response to major comment 1 above. Additionally, we have added a calibration to field data, which provides additional details not included in Zlotnik et al (2020). We appreciate the reviewer bringing this lack of clarification in the manuscript to our attention.

2.    Eq. A3 implicitly states that the absolute elevation of the frost table in the polygon rim is always equal to (or lower than) the thaw depth in the polygon center. Otherwise, there would have to be a condition, that kappa becomes zero (or very small) for z larger than the rim frost table elevation. This means that thaw depths in the centers are assumed significantly smaller than in the rims (due to the higher absolute surface elevation of the rim) in the model. While the authors write of a smaller "hydraulic conductive capacity" of the rims "due to a raised thaw table following the surface topography" (l. 135), this does not simply translate to a smaller kappa. In fact, all flowlines and the entire analysis change if the still frozen part of the polygon rim forms a threshold over which the water must drain. This for example means that the model is not really applicable early in the season, when thaw depths are low and naturally follow the microtopography. The authors need to present field measurements or other evaluations of the seasonal progression of thaw depths and associated microtopography that help evaluate in which situations the results can represent. It is important to know if the model is applicable 90% or only 10% of the time. They should also discuss in much more detail to what extent "general intuitive insights" (l. 105) from the model results can be transferred if the model assumptions are partly violated. Sect. 4.2 is not nearly enough and in my opinion omits the most critical limitations (see also next point).

We appreciate the reviewer bringing up this limitation of our model that readers should be appraised of. We agree that the raised frost table under the rim will alter flow lines. However, fundamental hydrology does not indicate that "all flowlines and the entire analysis change". An analogy can be drawn here with a partially penetrating well in an aquifer, where the effect on flowlines compared to a fully penetrating well dissipate quickly and are non-existent in the lateral direction after 1.5-2 times the aquifer thickness (Bear, 1979). In addition, the raised frost table will not change the entire analysis, but will simply warp the flowlines upwards in a localized region near the intersection of the frost table and the rim. So, while readers should be aware of this limitation, it does not negate our results or the general intuition that they provide. In order to ensure that readers are appraised of this, we have added the following discussion to Section 2.2:

"Note that while the raised thaw table under the rim will constrict flow and alter drainage pathways, hydrologic first principles indicate that this effect will be restricted to the region of the model near the rim. This is analogous to the effect of a partially penetrating well on flow in an aquifer, which dissipates quickly and is non-existent by a lateral distance of 1.5 to 2 times the aquifer thickness for isotropic aquifers (Bear, 1979). This effect will diminish with increasing

aspect ratio and decreasing anisotropy, and will not significantly alter the qualitative insights drawn from the overall drainage pathways identified in our analysis."

3. Anisotropy: The authors need to provide a clearer picture how and why anisotropy in hydraulic conductivities exists and what real-world cases different values of anisotropy represent, e.g. $K_r/K_z=100$. In particular the model representation of horizontal layers with highly different hydraulic conductivities, as it occurs for real-world-polygons, should be discussed. It looks like the simple model assumes horizontal and vertical hydraulic conductivities to be constant throughout the entire polygon center. This assumption should strongly determine the flowlines and thus the findings, but I am not at all convinced that it is a good representation of a real-world polygon center, where e.g. surface moss layers can have a strongly different hydraulic conductivity than mineral layers below. In the last point of their Conclusions, the authors explicitly describe layers with different hydraulic properties as a reason for the anisotropy, but this is not at all represented by the model ($K_r$ and $K_z$ in Eq. A2 have no depth dependency). Therefore, I do not think that the quantitative analysis is sound if there are layers with different hydraulic conductivities.

We agree, and state in the manuscript, that ice-wedge polygons are expected to have layers with variable hydraulic conductivities. As well, frost heave and cryoturbation are expected to provide some mixing of these layers in many cases. In short, the hydrologic properties of ice-wedge polygons are expected to be complex and have not been extensively measured to date. To this effect, we provide references to the existing limited information available on ice-wedge polygon anisotropy. We also agree, and clearly state in the manuscript, that our drainage patterns are based on effective properties, and that the actual drainage pattern for a given polygon will be altered by heterogeneous layering. While we agree that soil layering will alter the drainage patterns, readers are well appraised of this and can recognize that while the drainage patterns will deviate from our results, this does not negate the general qualitative intuition that our results provide regarding the relative effects of geometry and anisotropy on drainage patterns and timing.

Additionally, we have added a calibration extending over an entire thaw season that exposes the depth-dependent nature of the vertical hydraulic conductivity and that demonstrates the ability of the model to capture the effect of soil layers using effective hydraulic conductivities. The calibration identifies that in order to match water level observations, the effective vertical hydraulic conductivity must be a decreasing function of the thaw depth. So, while we fully recognize and agree with the reviewer that the model is unable to explicitly capture heterogeneity, the use of effective properties is able to capture the drainage dynamics of ice-wedge polygons. The following plot summarizing the calibration has been added to the manuscript:

[Figure]

*Figure 11 a) Progression of center water-level calibration. The (1) base calibration, (2) calibration with a precipitation multiplier, and (3) calibration with vertical hydraulic conductivity as a function of thaw depth (Kz =f(D)). The measured polygon-center and trough water levels are plotted for reference. Precipitation is plotted on the right y-axis. (b) Measured thaw depth and calibrated vertical hydraulic conductivity as a function of thaw depth.*

4.    The manuscript largely uses model-specific terminology which is hard to relate to real-world parameters, e.g. thaw depth and polygon diameters, in an intuitive way. It would make the manuscript more readable if the authors reword some of the statements to more plain language (see above for examples).

We appreciate the comment, and have re-evaluated the text and revised to "plain language" based on the examples above. For example, "aspect ratio" has been changed to descriptions of the physical geometry of the polygon and "anisotropy" changed to descriptions of the preferential flow direction where ever possible. Additionally, we have reevaluated the entire manuscript and made revisions to improve its overall readability.

Minor comments:

L. 82: How about the case that the thaw depth in the polygon rims is above the ground surface of the center, i.e. within the vertical interval of the pond? From my understanding, this situation is not represented by the model? In reality, there should be only negligible flow through the soil in the center. This could be an important situation early in summer.

Yes, we agree with the reviewer that limited drainage would occur under this scenario. We have added the following discussion to appraise the reader of this early season case in section 2.2:

"It is conceivable that early in the thaw season the permafrost table under the rim could extend above the center ground surface within the vertical interval of the center pond. In this case, there will likely be very little drainage occurring associated with a very small discharge conductance."

L. 97: but that also implies a depth dependence of anisotropy, which does not seem to be accounted for in the model. See major comments.

The reviewer is completely correct, and we now provide quantitative evidence to support this through a season-long calibration to field observations. The calibration clearly illustrates the depth dependence of the effective vertical hydraulic conductivity consistent with many observations of polygon soil layering. Please refer to our response to major comment 3 above and Figure 11.

L. 105: I have the impression that the limitations of the model are quite severe (see major comments), so the "general intuitive insights" might not be applicable for many relevant cases. It is important to discuss and present this in more detail.

We appreciate the reviewer's skepticism and insistence that the model limitations be fully explored. This is accomplished in two ways in the manuscript: (1) lengthy discussions of model limitations and their implications and (2) a calibration to field observations demonstrating the model's ability to capture ice-wedge polygon drainage dynamics.

Considering (1), we mention and discuss the limitations of the model in many places with the following examples:

In the abstract:

"Our results, based on parametric investigation of idealized scenarios, provide a baseline for further research considering the geometric and hydraulic complexities of ice-wedge polygons."

The Introduction:

"Although the simplifications of the model may limit its applicability to some scenarios, they allow general intuitive insights to be drawn which would be obfuscated without them. The findings here provide a basis to quantify and understand deviations from our idealized scenarios."

The Methods Section:

"We have neglected the effects of evaporation and precipitation as they will not affect the drainage patterns we present (based on non-dimensional heads) and their effect on drainage timing (based on non-dimensional depletion curves) is straightforward, shifting the nondimensional exponential drainage curve upwards or downwards. In other words, using non-dimensional variables is a powerful approach to gain intuition into the fundamentals of inundated ice-wedge polygon drainage irrespective of variable magnitude."

And all of Section 4.2 "Model limitations".

For (2), we have added a calibration to field observations to test the ability of the model to capture ice-wedge polygon drainage dynamics over an entire thaw season. The calibration indicates the minimum refinements necessary to our parsimonious model to allow it to match observed ice-wedge polygon water levels. Please refer to our response to major comment 3 and Figure 11 above.

L. 185: Is such a high range for the anisotropy reasonable (what kind of material would the outer limits correspond to)? See major comment on the layering.

The reviewer's comment brings to our attention that a more detailed discussion of effective properties and their physical interpretation in layered media is needed in the text. While this was already alluded to in the Introduction, we have now added the following discussion to clarify:

"A physical interpretation of our selected values of anisotropy can be obtained by considering that ice-wedge polygon soils are typically layered and that the horizontal and vertical hydraulic conductivities can therefore be considered as *effective* properties. As such, the effective horizontal hydraulic conductivity captures parallel flow dominated by the higher conductivity layers and the effective vertical hydraulic conductivity captures flow in series across multiple layers and is dominated by the lower conductivity layers. Therefore, an anisotropy of 100 would *effectively* capture layers with 2 orders of magnitude difference in hydraulic conductivity, while an anisotropy of 0.1 would capture the hypothetical scenario where vertical cracks or burrows result in preferential vertical flow. Given the current lack of direct measurements of ice-wedge polygon anisotropy, we cover a broad range of possible scenarios."

L. 190/Fig. 6: See major comment on rim hydraulic conductivity and frost table. The seasonal deepening of the frost table in the rim which likely is a major control for drainage from the polygon does not seem to be accounted for in the model. Modeled depletion curves extend over periods of a month and more, for which this thaw progression is highly relevant.

We recognize the reviewer's concern that while our sensitivity analysis uses a model with a fixed frost table, during the drainage time frame, this may only apply at late seasons (e.g., see Fig. 11 above). It should be noted that the purpose of the sensitivity analysis was to remove as many variables as possible to focus on geometry and anisotropy. This is a generally accepted scientific approach that allows factors to be isolated that we believe many readers will appreciate. We provide the following discussion to appraise the reader of this strategy, its limitations, and guide their interpretation of the results in Section 2.2:

"In practice, the water level in troughs ($H_t$ in equations 1 and 3) will vary throughout the thaw season, affecting the magnitude of heads in the soil of the polygon center and drainage times. As the non-dimensional heads are relative to $H_t$ (refer to 220 equation A7), its value does not affect our relative comparisons of drainage patterns (which are based on non-dimensional heads that are normalized from 0 to 1). The value of $H_t$ will affect our comparisons of drainage time, but in a systematic, interpretable manner. For example, a higher $H_t$ will compress the exponential curve defined by equation 3 upwards, while a lower $H_t$ will stretch the exponential curve downwards. In cases where $H_t$ is below the polygon-center ground surface, the solution is valid until $H_c$ reaches the ground surface, at which time the ponded center has completely drained. Therefore, to isolate our analysis to aspect ratio and anisotropy, we have set $H_t$ in all cases equal to the polygon-center ground surface."

L. 245: the term "ridgeline" could create confusion with "polygon rims".

Good point, we now refer to it as a "curved feature".

L. 336: It would be nice to state some of this in more intuitive language, e.g. "for a given thaw depth advective heat transport to the thaw front in the polygon rims is higher for large polygons".

Great suggestion, the sentence now reads:

"For a given thaw depth, advective heat transport will be more focused near the rim for larger polygons, and may result in enhanced ice-wedge degradation (Wright et al., 2009)."

---

## Author Response (AR2)

Response to Report #1, Anonymous Referee #2

The authors have added model calibration/validation to the manuscript, which is clearly novel compared to the previous Zlotnik et al. publication. This addresses and at least partly resolves the most concerning issue I had with the last version.

We appreciate the reviewer taking the time to provide another detailed review of our manuscript. The major improvements to the manuscript include:

- Reorganization sections to begin with the calibration and follow with the sensitivity analysis. This significantly improves the flow of the manuscript.
- Details on the potential errors in the driving data are now provided and discussed. We state the resolution and/or accuracy of the measurements. We also provide standard errors for the parameter estimates from the calibration informing the reader how well constrained each parameter is and discuss their implications.
- It is now clearly stated how the current manuscript extends and expands on the results from Zlotnik et al. (2020).
- The role of the precipitation multiplier as not only a means to deal with rain gauge under catch, but also as a way to account for the inevitable runoff from rims into the center pond is clearly stated.
- It is now clearly explained that continuous local ET measurements are not available for our site over the 2013 thaw season. It is also explained that a large scale ET product is sufficient for our purposes given the continuous (albeit diurnal) low magnitude of ET, in contrast to the episodic precipitation record where local measurements are crucial.

Detailed responses to reviewer comments are provided below.

While the novelty aspect is improved, the manuscript needs restructuring before it can be considered for publication. The new paragraphs on calibration/validation have largely been inserted in the already established structure of the previous version, but this does not fit in my opinion. For example, Sects. 3.1 - 3.3 elaborate on the model sensitivity, but then Sect. 3.4 states that the simple model (which is the basis for 3.1-3.3) does not fit the measured data, but needs to be modified as in case 3 (I like the clear and unambiguous language on the model evaluation in 3.4!). This can be confusing for the reader, although most (or maybe even all) of the analysis in 3.1-3.3 still holds.

As the reviewer suspects, all the sensitivity analyses still hold because they are snapshots in time using nondimensional parameters. Therefore, the sensitivity analyses are not affected by the precipitation factor or the thaw-depth dependent Kz identified in the calibration. This is now clearly stated in section 3.1 "Calibration to field measurements:

"The calibration verifies that the model is able to capture ice-wedge polygon drainage characteristics. In the next sections, we perform sensitivity analyses using non-dimensional forms of this verified analytical solution to gain insights into ice-wedge polygon drainage

characteristics. The use of non-dimensional solution snapshots eliminates the need to consider the precipitation multiplier and thaw-depth dependent vertical hydraulic conductivity explicitly. Instead, their effects are implicit in the relative differences between snapshots."

So my suggestions for restructuring and further analysis are:
1. Make it clearer in the Sections on study design that calibration/validation of the Zlotnik-model is one of the primary goals of the study. This part clearly goes beyond the Zlotnik et al.-paper, and that is what the revised manuscript must demonstrate. The authors write in their reply that the manuscript "rigorously confirms, fully fleshes out, and expands on" Zoltnik et al., which could be seen as a positive formulation for "the conclusions are essentially the same". So the authors should extend and deepen the new part and provide more extensive information how the cal/val model runs are set up. Also mention the spatial and time resolution and known or probable uncertainties/ error characteristics of the driving data sets (e.g. for evaporation) and how this affects the calibration procedure.

We agree that the paper is better organized in the way that the reviewer suggests, and we have switched the order throughout the manuscript. We are thankful for the suggestion. We also appreciate that the reviewer recognizes that the calibration/validation "clearly goes beyond the Zlotnik et al. paper". In Section 2.2 "Calibration approach", we provide a complete description of how the cal/val model runs are set up. We state that we use a Levenberg-Marquardt approach and define the objective function and the parameters and meta-parameters for each calibration case. This provides a comprehensive description of the cal/val model setup for readers.

We do not agree with the reviewer's characterization that "the conclusions are essentially the same" from the sensitivity analysis in this manuscript as what is in Zlotnik et al. (2020). Aside from the new information provided in the calibration, Figures 4-11 greatly expand on the preliminary scoping analysis in Zlotnik et al. (2020) providing new insights and information. This is clearly stated in the abstract:

"We also provide a comprehensive investigation of the effect of polygon aspect ratio and anisotropy on drainage timing and patterns expanding on previously published research. Our results indicate that polygons with large aspect ratios and high anisotropy will have the most distributed drainage, while polygons with large aspect ratios and low anisotropy will have their drainage most focused near their periphery and will drain most slowly. Polygons with small aspect ratios and high anisotropy will drain most quickly."

Also, it is readily apparent in Figures 7 and 10 that new information not provided in Zlotnik et al. (2020) is being presented in global sensitivity analyses of geometry and anisotropy which provides detailed non-linear nuances in drainage patterns due to combinations of geometry of anisotropy. The preliminary scoping analyses of Zlotnik et al. (2020) did not provide this information. This is indicated and elaborated on in the conclusions:

"We provide rigorous confirmation that the majority of drainage from inundated ice-wedge polygon centers occurs along an annular region along their radial periphery; however, polygon geometry and hydraulic conductivity anisotropy significantly impact the drainage pathways, as originally postulated by Zlotnik et al. (2020)."

The conclusions continue providing details on the global sensitivity insights derived from our manuscript that are not provided in Zlotnik et al. (2020).

The probable uncertainties/error characteristics of the driving data sets are now fully described in Section 2.4 "Acquisition of field data used in calibration" where the known accuracy or resolution of datasets are stated. We also now describe in detail why we used a large-scale ET product and the implications that has on our analysis:

"Due to a lack of continuous local evapotranspiration measurements, we obtained evapotranspiration data from NASA's Global Land Data Assimilation System (GLDAS) (Rodell et al., 2004). Given the continuous (albeit diurnally fluctuating), low magnitude evapotranspiration signal, its effect on our calibration is relatively insignificant compared to the sporadic precipitation events that drive large scale fluctuations in water levels. Therefore, in lieu of local evapotranspiration measurements, the GLDAS evapotranspiration is deemed sufficient for our purposes here."

The probable uncertainties/error characteristics of the driving data sets are also now discussed in section 4.2 "Calibration implications":

"The water level (Liljedahl and Wilson, 2016) and temperature measurements (Romanovsky et al., 2017) are well constrained with high degrees of resolution and accuracy. While the resolution of precipitation measurements is high (0.1~mm), there is the potential for under catch during windy precipitation events. More importantly with regard to ponded water levels, the precipitation measurements do not account for the runoff of water during precipitation events from rims into the polygon center, resulting in measured precipitation less than increases in pond water levels. This uncertainty is accounted for through the calibration of a precipitation multiplier, which effectively captures the effect of runoff from the polygon *watershed* into the center pond along with any potential under catch during windy precipitation events."

2. Start Sect. 3 results with the cal/val section (now 3.4), and based on the findings motivate the sensitivity analyses 3.1-3.3 and why they are still meaningful in the light of the cal/val findings. I am also missing a closer analysis of case 3 and the sensitivity of the resulting parameters. I guess equifinality could be a significant problem here, so how well-defined is the minimum in RMSE? Are there other parameter combinations that result in a similar RMSE? Are some of the parameters better constrained by the analysis than others? For example is the value for the minimum hydraulic conductivity (about 5e-8 m/sec which is a reasonable value for poorly permeable silt) well-defined, or would a whole range of values, from e.g. 1e-6 to 1e-9 m/sec, provide a similar performance? How do uncertainties in input parameters, in particular

evaporation (which also leads to a drop in water levels) affect the calibration? The authors seem to use a large-scale ET product (which integrates over rims and centers?), but the model needs precipitation for a wet polygon center. Have the authors tried an evaporation multiplier?

The results section now begins with the cal/val section. The sensitivity analysis is now motivated at the end of the cal/val section as:

"The calibration verifies that the model is able to capture ice-wedge polygon drainage characteristics. In the next sections, we perform sensitivity analyses using non-dimensional forms of this verified analytical solution to gain insights into ice-wedge polygon drainage characteristics. The use of non-dimensional solution snapshots eliminates the need to consider the precipitation multiplier and thaw-depth dependent vertical hydraulic conductivity explicitly. Instead, their effects are implicit in the relative differences between snapshots."

In order to address the question of "equifinality", standard errors are now provided for each parameter in Table 1. The following discussion has been added to Section 3.1 as well:

"The standard errors of the calibrated parameters for calibration case 3 listed in Table 1 indicate how well constrained the parameters are by the calibration. It is apparent that the hydraulic conductivities (horizontal and minimum and maximum vertical) are not well constrained with relatively large standard errors. These parameters (or their meta-parameters) also have large covariances with each other indicating their correlated effect on the model. However, despite the lack of constraint of these parameters due to their correlated effect on the model, the calibration does identify reasonable values for them. The standard errors of the discharge conductance, initial polygon-center water level, and precipitation multiplier indicates that they are well constrained by the calibration."

A large-scale ET product had to be used as continuous local ET measurements are not available at the site in 2013. Since ET is a relatively small, continuous (albeit diurnal) driver for our model, its affect is much less significant than sporadic precipitation events (refer to Figure 3). So, while ET does lead to a significant loss of water over the entire thaw season, it is not crucial in capturing fluctuations in water levels in the same way that precipitation events are. Therefore, the use of the large-scale ET product does not significantly affect the conclusions drawn from the calibration. Along the same lines, we do not feel that exploring an evaporation multiplier would provide any additional insights into polygon drainage in this case. We have added the following text in Section2.4 "Acquisition of field data used in calibration" to clarify this point for the reader:

"Given the continuous (albeit diurnally fluctuating), low magnitude evapotranspiration signal, its effect on our calibration is relatively insignificant compared to the sporadic precipitation events that drive large scale fluctuations in water levels. Therefore, in lieu of local evapotranspiration measurements, the GLDAS evapotranspiration is deemed sufficient for our purposes here."

3. Restructure the Discussion accordingly, exactly same issue as for the Results section! Furthermore, the authors should discuss how clear model deficiencies (as raised in the earlier review report and confirmed in the author's reply) could affect the calibration and the hereof derived conclusions. In l. 521, the authors write rather boldly that the model "helps identify factors which need to be considered by any hydrologic model to simulate drainage from an inundated polygon center". Considering that the model does not account for obvious factors, e.g. different thaw depths in the rim and the center, it is important to substantiate this statement with a clear analysis of the model uncertainties and an assessment of how reasonable the resulting parameters are. A few parameters are discussed in Results, e.g. Kr, but this should be expanded to all parameters in Table 1. As an example, the precipitation multiplier of >2 is casually explained by undercatch, but is >100% undercatch possible/reasonable for summer rainfall (not winter snowfall) for the particular rain gauge type that it was measured with? The publication Pollock et al. (2018) cited by the authors reports an undercatch of 23% and discusses the effect of different rain gauge types in detail. If an undercatch of 100% was not reasonable, this would be an indication that the model produces the right results for the wrong reasons (also see my comment on an evaporation multiplier above), and this would need to be analyzed, discussed, and the conclusions adapted accordingly.

The discussion section has been reorganized accordingly. A more complete discussion of parameters is now included in the "Calibration implications" section:

"Due to covariance in the effects of the hydraulic conductivity parameters on water levels, the hydraulic conductivity parameters are loosely constrained by the calibration (based on local sensitivities). This indicates the importance of constraining the hydraulic conductivity parameters with field measurements if possible (field measurements are not available in our case). However, despite relatively large standard errors in the hydraulic conductivity parameter estimates, the calibration identifies physically realistic values. It should also be noted that despite vertical hydraulic conductivity being loosely constrained in the final calibration (calibration case 3), the model was unable to match water levels    with a constant vertical hydraulic conductivity (calibration case 2). This indicates the importance of the thaw-depth dependent vertical hydraulic conductivity in the model. The other parameters (discharge conductance, initial polygon-center water level, and precipitation multiplier) are all well constrained by the analysis."

We have also added a paragraph in the "Calibration implications" section addressing data uncertainty:

"The water level (Liljedahl and Wilson, 2016) and temperature measurements (Romanovsky et al., 2017) are well constrained with high degrees of resolution and accuracy. While the resolution of precipitation measurements is high (0.1~mm), there is the potential for under catch during windy precipitation events. More importantly with regard to ponded water levels, the precipitation measurements do not account for the runoff of water during precipitation

events from rims into the polygon center, resulting in measured precipitation less than increases in pond water levels. This uncertainty is accounted for through the calibration of a precipitation multiplier, which effectively captures the effect of runoff from the polygon *watershed* into the center pond along with any potential under catch during windy precipitation events."

We agree that an under catch of >100% is not likely. However, the precipitation multiplier is not "casually explained by under catch" in the manuscript. In several places in the manuscript, we explain that the precipitation multiplier not only accounts for potential rain gauge under catch, but also accounts for the fact the pond in the center of the polygon will collect precipitation runoff from the surrounding rims and other high ground within the polygon center. The area of rims and other high ground can be a significant portion of the polygon *watershed*, resulting in the increase in the center pond height being much larger than the precipitation measured by a rain gauge. This was clearly stated in the previous manuscript and has now been elaborated on (see paragraph above) and clarified even further in the abstract:

"…accounts for runoff from rims into the ice-wedge polygon pond during precipitation events and possible rain gauge under catch…"

Section 2.2 "Calibration approach":

"In calibration case 2, we add a precipitation multiplier $M_P$ to case 1, where $\hat{P} = M_P P$ and $\hat{P}$ is the augmented precipitation accounting for runoff from microtopographic highs and potential rain gauge under catch."

Section 3.1 "Calibration to field measurements":

"Considering that precipitation will runoff from rims and collect in the polygon-center pond and that rain gauges may have under catch issues…"

And Section 4.1 "Calibration implications":

"The first refinement is a precipitation multiplier and is based on a simple mass balance indicating that the measured precipitation cannot account for the total increase in ponded water levels after precipitation events. The precipitation multiplier accounts for the fact that precipitation will run-off from the rims into the center pond resulting in a larger increase in ponded water than rain gauge precipitation. The precipitation multiplier also accounts for any potential rain gauge under catch."

These multiple discussions throughout the manuscript will clarify the role of the precipitation multiplier for readers.

---

## Author Response (AR3)

**Response to review by Editor**

Dear Dr. Langer,

We are pleased that you see no major obstacles to publication and that you provide feedback for us to response to. Please find our responses below in blue.

**Editor Decision: Publish subject to minor revisions (review by editor)** (15 Jul 2021) by Moritz Langer
Comments to the Author:
Dear authors,

Thank you for your careful consideration of the reviewer's comments and suggestions. I conclude that all important issues have been adequately addressed in the revised manuscript, so I see no major obstacle to publication. However, please note the following technical comment and two questions regarding the conclusions from my side:

In section 2.2 and other sections, there seem to be technical problems in compiling the citations.

Yes, I had difficulty with some of the references in the mark-up version of our revisions. Instead of taking the time to put the actual citations in those cases, I simply left them as the bibtex label. My apologies for this confusion. This is only an issue in the mark-up version. It can be verified that they are correct in the actual manuscript.

l. 695ff: Would this general statement hold if small and deep polygons had higher flow resistance in their rims?

This is a good question. I assume that you mean if small, deep polygons characteristically had higher flow resistance in their rims than wide, shallow polygons? I suppose that perhaps that could be the case, although I'm not aware of any research indicating that. Given the lack of information regarding flow resistance under rims in general, not to mention differences between small and deep vs wide and shallow polygons, our sensitivity analysis assumes constant flow resistance in their rims.

l698ff: This conclusion seems a bit contradictory to the previous one. It could be expected that small and deep polygons would have a larger fraction of cryoturbated area/volume, while wide and shallow polygons should have a larger area unaffected by ice wedge expansion leading to a more layered stratigraphy. Would it be possible that the two effects cancel each other out in reality?

Yes! Great point! This is the whole point of Figure 10 (and analogously Figure 7 for drainage pattern). We stress in the manuscript that the combination of anisotropy and aspect ratio need to be considered to determine the drainage pattern and timing. To clarify, we have added that

in the former case, we are referring to when anisotropy is held constant and in the latter case when aspect ratio is held constant.

Best regards
Moritz Langer